# Slab control on the mega-sized North Pacific ultra-low velocity zone

Jiewen Li [1,2], Daoyuan Sun [1,2 ✉] & Dan J. Bower [3]

Ultra-low velocity zones (ULVZs) are localized small-scale patches with extreme physical properties at the core-mantle boundary that often gather at the margins of Large Low Velocity Provinces (LLVPs). Recent studies have discovered several mega-sized ULVZs with a lateral dimension of ~900 km. However, the detailed structures and physical properties of these ULVZs and their relationship to LLVP edges are not well constrained and their formation mechanisms are poorly understood. Here, we break the degeneracy between the size and velocity perturbation of a ULVZ using two orthogonal seismic ray paths, and thereby discover a mega-sized ULVZ at the northern edge of the Pacific LLVP. The ULVZ is almost double the size of a previously imaged ULVZ in this region, but with half of the shear velocity reduction. This mega-sized ULVZ has accumulated due to stable mantle flow converging at the LLVP edge driven by slab-debris in the lower mantle. Such flow also develops the subvertical north-tilting edge of the Pacific LLVP.

[1] Laboratory of Seismology and Physics of Earth's Interior, School of Earth and Space Sciences, University of Science and Technology of China, Hefei, Anhui 230026, China. [2] CAS Center for Excellence in Comparative Planetology, China, Hefei, Anhui 233500, China. [3] Center for Space and Habitability, University of Bern, Gesellschaftsstrasse 6, 3012 Bern, Switzerland. ✉email: sdy2014@ustc.edu.cn

The Earth's lowermost mantle is dominated by two Large Low-Velocity Provinces (LLVPs) beneath Africa and the mid-Pacific. While tomographic models show coherent images of LLVPs at large scale[1–6], discrepancies exist among different models regarding the fine-scale structures and detailed boundaries. There is no consensus about the origin of LLVPs but they are probably both thermally and chemically distinct structures[7–10]. Along the LLVP margins, thin patches (10–100 km in height) of ultra-low velocity zones (ULVZs), characterized by a reduction of 5–25% in compressional and 10–45% in shear velocity, are widely distributed at the core-mantle boundary (CMB)[11,12]. Different mechanisms have been proposed to explain the drastic velocity drops of ULVZs, including iron-enriched assemblages[13], partial melting[14], hydrogen-bearing iron peroxide[15], and silicate sediments from the core[16]. Temporal and spatial correlations between hotspots and reconstructed large igneous provinces (LIPs) with LLVPs[17] are suggested. Also, many ULVZs are likely to distribute over LLVP margins[11]. Thus, LLVPs and ULVZs may be reservoirs of recycled oceanic crust[18] and/or primordial materials[19,20], which can be viscously entrained by mantle plumes to generate a wide range of geochemical signatures in oceanic island basalts[21].

Assessing the geometries and physical properties of LLVPs and ULVZs requires constraints from seismic waves sampling these structures. The LLVP beneath southern Africa is well studied with ideal seismic phases, such as SKS and Sdiff, which sample the edge of the LLVP and determine its boundary sharpness[8,22]. However, because of limited data coverage, most studies of the Pacific LLVP focus on characterizing its edge structure[23,24] and in particular ULVZs[25–30]. Cottaar and Romanowicz[25] propose a large-sized ULVZ near Hawaii (referred to as the Hawaiian mega-sized (HM) ULVZ hereafter) at the northern edge of the Pacific LLVP (Fig. 1), which is ~910 km in diameter, 20 km in height, and −20% in shear velocity perturbation ($\delta V_S$), by mainly modeling strong Sdiff postcursors along a west-east (W-E) profile. Now with USArray moved to Alaska in recent years, we have an unprecedented opportunity to study the northern edge of the Pacific LLVP in a north-south (N-S) direction (Fig. 1) to resolve strong trade-offs among location, velocity perturbation, and size of the north Pacific mega-sized ULVZ. Other identified mega-sized ULVZs correlate with hot spots directly beneath Iceland[31] and Samoa[32,33]. In contrast, the mega-sized ULVZ at the northern edge of the Pacific LLVP does not overlap with the Hawaiian hotspot geographically. Thus, resolving the boundaries of ULVZs and LLVPs is essential to uncover if mega-sized ULVZs are the root of deep mantle plumes.

In this study, we discover a mega-sized ULVZ at the northern edge of the Pacific LLVP and determine that its shear velocity perturbation is −10% and the size is ~1500 × 900 km. Combined with the imaged subvertical north-tilting edge of the Pacific LLVP, we demonstrate that a stable mantle flow converging at the LLVP edge can significantly affect the dynamic evolution of the ULVZ and LLVP.

## Results

**Height of the Pacific LLVP.** For 13 events with simple source time function between 2016 and 2020 (Fig. 1, Supplementary Table 1) in the Tonga-Fiji-Kermadec region recorded by stations in Alaska, we measure the ScS-S differential travel time residuals ($\delta t_{ScS-S}$) relative to IASP91[34] with epicentral distances smaller than 85°, at which S and ScS are well separated. For event B, we measure the $\delta t_{ScS-S}$ with epicentral distance extending to 90° due to the high-quality data (Fig. 2c). Interestingly, $\delta t_{ScS-S}$ correlates positively with ScS travel time residuals ($\delta t_{ScS}$) and negatively with S travel time residuals ($\delta t_S$) (Fig. 2a, b, Supplementary

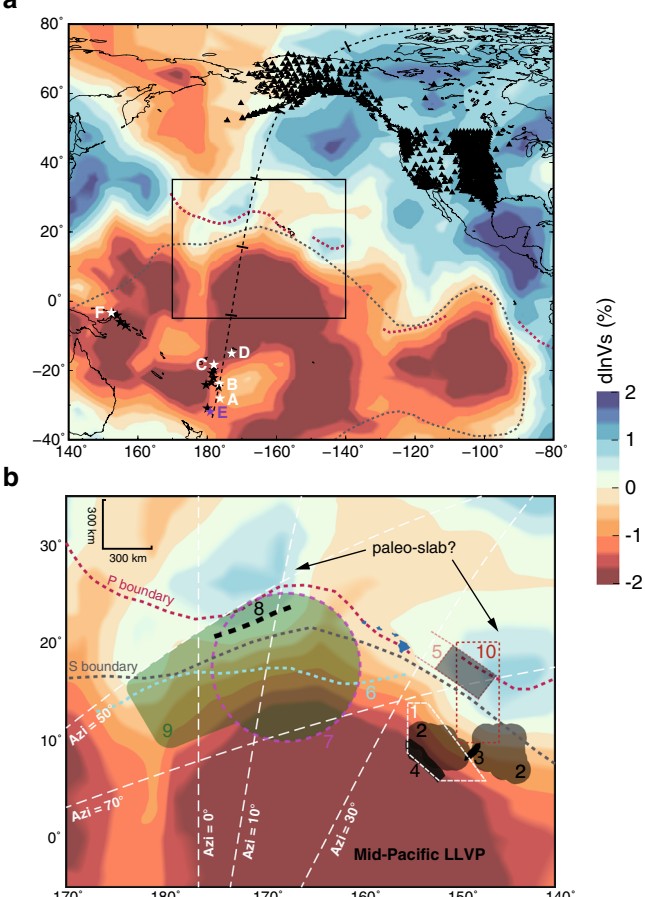

**Fig. 1 Locations of events and stations used in this study. a** Five events (A, B, C, D and F, white stars) and stations (black triangles) are used to model the seismic structure of the Pacific Large Low Velocity Province (LLVP) and ultra-low velocity zone (ULVZ) in detail. Event E (purple star) is used to correct S travel times (see text for details). A cross-section along the black dashed line with 20° tick interval is displayed in Fig. 3a. **b** Enlarged view of the inset box in **a**. The northern shear (S) wave[24] and compressional (P) wave boundaries[35] of the Pacific LLVP are indicated by grey and red dashed lines, respectively. The white dashed polygon 1 indicates an area of strongly varied D″ structure[55]. The shaded regions with corresponding fonts in color are inferred ULVZs: 2[54], 3[27], 4[26,55,62–65], and 5[29]. The cyan dashed line 6 displays the ULVZ boundary inferred from a PKP study[28]. The magenta dashed circle 7 represents the Hawaiian mega-sized (HM) ULVZ[25]. The newly mapped LLVP boundary and the North Pacific (NP) ULVZ are marked with heavy black dashed line 8 and green shadow region 9, respectively. Red box 10 marks the base of the modeled Hawaiian plume[56]. The white dashed lines mark different azimuths for events B and F. The background tomography image is from the GyPSuM model[5] at the core-mantle boundary. Note there are high-velocity regions, which are related to the paleo-slab, situated at the northern and northeastern edge of the Pacific LLVP and D″ discontinuities are observed in these two regions[24,29].

Fig. 1), which suggests that both ScS and S contribute to $\delta t_{ScS-S}$. At azimuths of 0°−10°, $\delta t_{ScS-S}$ of event B advances by ~5 s from distance 75° to 90° (Fig. 2c), which is typically attributed to the change of $\delta t_{ScS}$. However, $\delta t_{ScS}$ appears constant along the distance profile. In contrast, $\delta t_S$ has a remarkable change up to 8 s with increasing distance, suggesting a strong anomaly in the lower mantle sampled by S (Fig. 3a).

Four events (events A–D in Fig. 1) in a N-S corridor with azimuth range of 0°–10° are selected to further examine $\delta t_S$

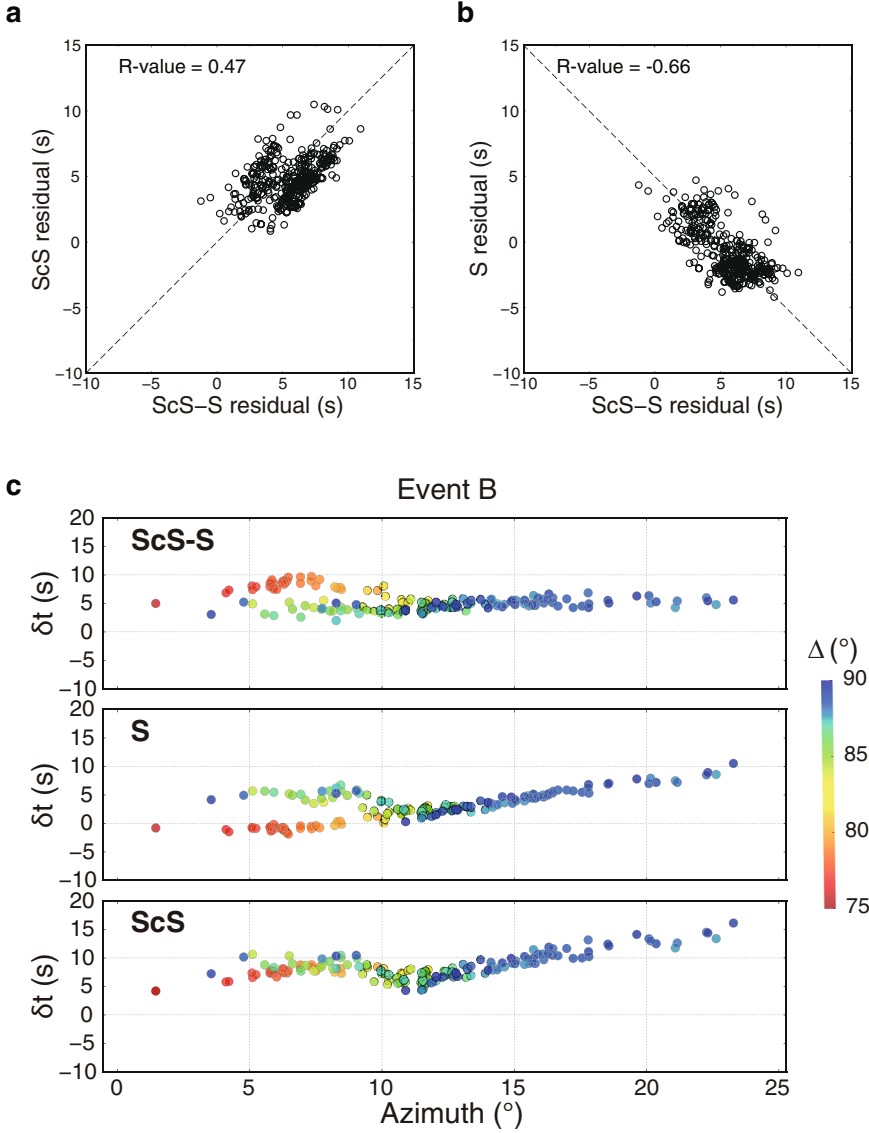

**Fig. 2 Relationship between travel time residuals of S and ScS.** The S/ScS travel time residuals ($\delta t_S$/$\delta t_{ScS}$) and differential travel time residuals between ScS and S ($\delta t_{ScS-S}$) are measured along the north-south profile in Fig. 1a. **a**–**b** Results after correction for GyPSuM model[5] at a depth of 0−2000 km. Travel time residuals are corrected to a common station UNV to remove possible errors of event origin time and location. Pearson's R-values quantify the correlations. **c** Azimuthal variation of travel time residuals ($\delta t_{ScS-S}$, $\delta t_S$, and $\delta t_{ScS}$) of event B with respect to the IASP91 velocity model[34]. The different color denotes the different epicentral distance. Note the large delay of S by up to 8 s from distance 75° to 90° at azimuths of 0°−10°.

(Fig. 3c). For the southern event A, the change of $\delta t_S$ is up to 5 s from 80° to 90°. Event B displays a sharp 8 s change of $\delta t_S$ from 80° to 87°. Event C is similar to event B but with a smaller $\delta t_S$ jump of 6 s from 75° to 82°. For the northernmost event D, the change of $\delta t_S$ drops to 5 s. To evaluate the effects of shallow structure on travel time anomalies, we use the travel time anomaly of SKS ($\delta t_{SKS}$) of event E (Fig. 3a, Supplementary Fig. 2) with ray sampling away from the northern edge of the LLVP to correct for upper mantle structure. After correction, the $\delta t_S$ variation is smaller but still has a ~6 s and ~5 s delay for event B and C (Fig. 3c), respectively. We also test using SS as the reference to correct for possible shallow effects, which displays a similar trend as using SKS correction (see Methods in detail, Supplementary Fig. 2). The consistency between corrections using different reference phases is true for all four events (Supplementary Fig. 2c), which further confirms that shallow mantle structure beneath stations or events has little effect on the corrected $\delta t_S$. Instead, the delay of $\delta t_S$ is mainly controlled by S

with its ray paths sampling a strong lower mantle structure, i.e. the Pacific LLVP.

When the S wave begins to encounter the top of the LLVP, $\delta t_S$ starts to increase. Thus, the distance of the initial increase of $\delta t_S$ provides a robust and tight constraint on the height of the Pacific LLVP (Supplementary Fig. 3). With events moving to the north, the distance at which $\delta t_S$ starts to increase becomes smaller (Fig. 3c), indicating the Pacific LLVP has a higher relief towards its northern edge. Based on these constraints, we outline a LLVP model that has the height varying from 720 km in the south to 920 km in the north (Fig. 3a), which is double the height in a previous model[24]. Then, we explore a series of models with different $\delta V_S$ in the LLVP and examine how $\delta t_S$ of different events are affected. We find that a LLVP with $\delta V_S$ varying gradually from -2% at the top to -4% at 300 km above the CMB and tilting to the north can best match the corrected $\delta t_S$ for events A–D on the N-S profile (Fig. 3b). For event A, the S wave consistently samples inside the LLVP so there is no obvious $\delta t_S$

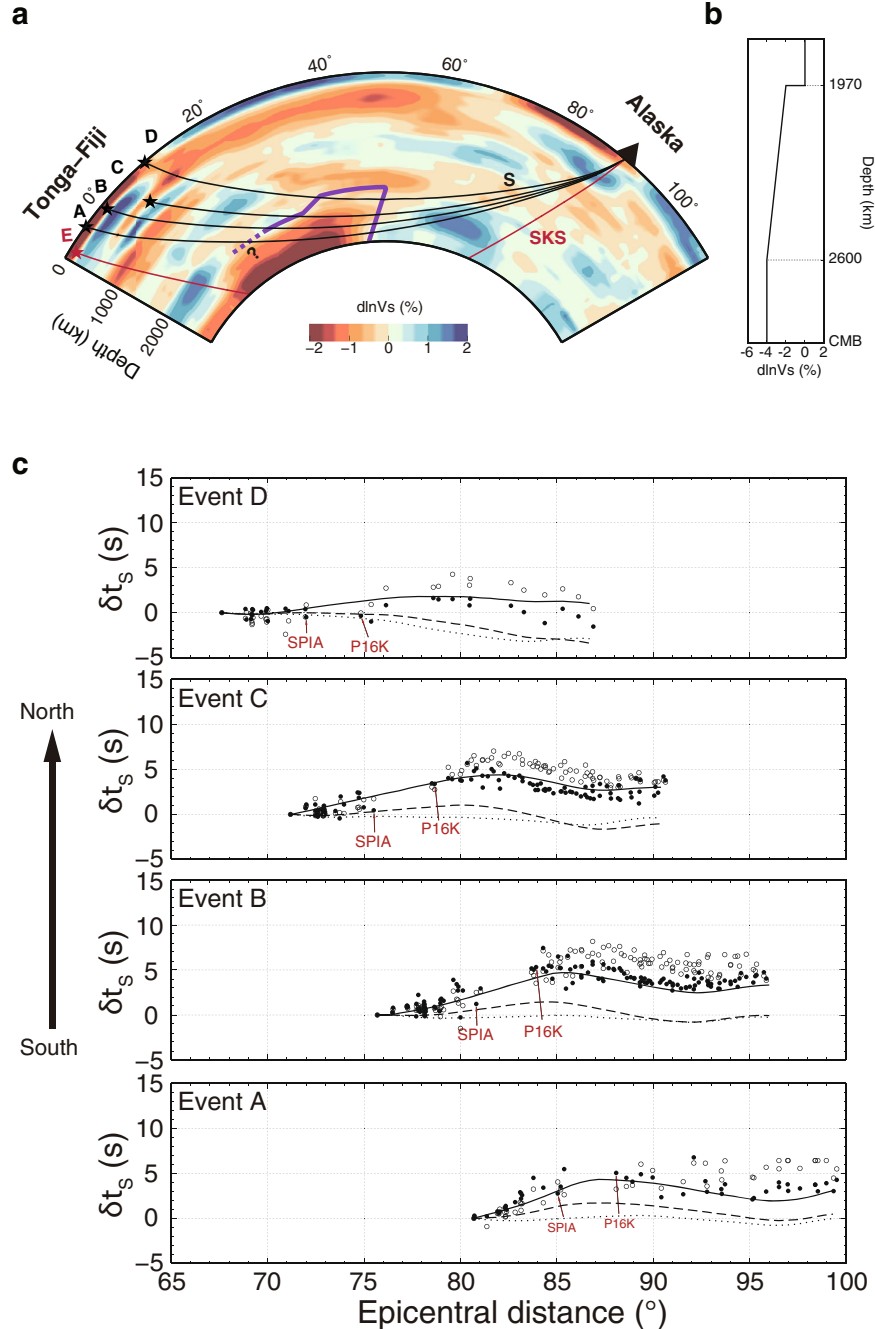

**Fig. 3 The northern edge of the Pacific Large Low-Velocity Province. a** Depth cross-section from Tonga-Fiji to Alaska in the GyPSuM model[5]. The ray paths of S (black) and SKS (red) are plotted at one common station from different events (stars). The purple line outlines our inferred Large Low-Velocity Province (LLVP) while the southernmost portion (purple dashed line) is not well constrained. **b** Shear wave velocity perturbation with depth relative to the IASP91 velocity model[34] in our inferred LLVP. **c** S travel time residuals (open circles) relative to the IASP91 velocity model[34] and after correction (solid circles) using SKS travel time residuals in Supplementary Fig. 2, along with predictions from the GyPSuM model (dashed line), GyPSuM at a depth of 0−2000 km (dotted line) and the new LLVP model (solid line) in **a**. Events are arranged from south to north. Two common stations (SPIA and P16K) are highlighted to show different S travel time trends of events A−D. Data are selected in an azimuth range of 0°−10°.

jump. In contrast, the S ray paths of events B–D sample above the LLVP at smaller distance and $\delta t_S$ increases quickly when S samples more LLVP with increasing distance. A velocity drop of 2% across the top of the LLVP is needed to match the rapid change of $\delta t_S$, and the gradual change of velocity perturbation at deeper depth reproduces the travel time anomalies for all events. To match the travel time of S for event A at larger distance, a uniform $\delta V_S$ of −4% is necessary in the lowermost 300 km of the mantle.

**Northern wall of the Pacific LLVP.** The $\delta t_S$ from the W-E section for events in the Solomon Islands (Supplementary Table 1) recorded by stations at North America provides extra constraints on the basal structure of the LLVP[24,35] in an orthogonal direction. From north to south, $\delta t_S$ slows up to 5 s (Fig. 4a, Supplementary Fig. 4). The largest gradient of $\delta t_S$ separating the S-wave turning points in the lowermost mantle is defined as the boundary of the Pacific LLVP. When $\delta t_S$ measured for the W-E profile in Fig. 4a are projected on a N-S profile (Fig. 4b) at their

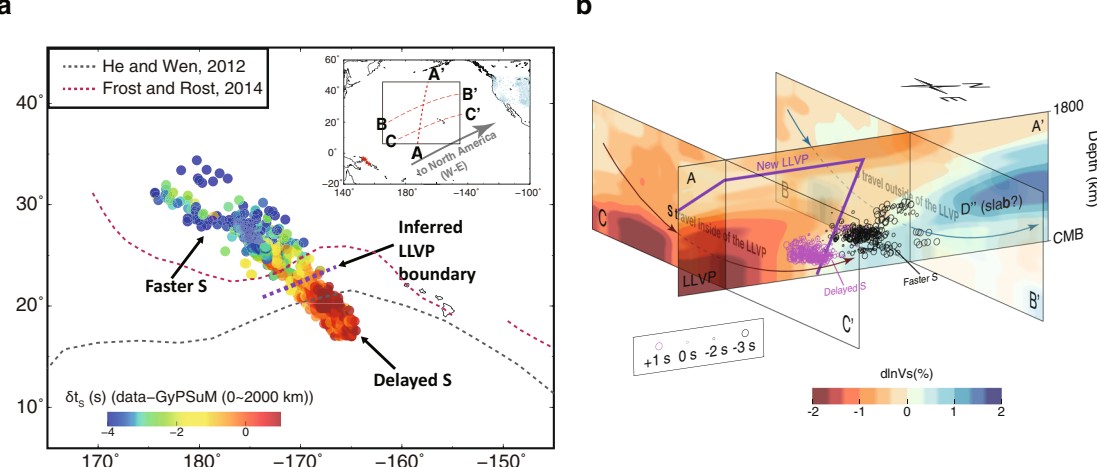

**Fig. 4 S travel time anomaly from West-East profile. a** Enlarged view marked with black box in inset showing S travel time residuals for events in Solomon Islands (red stars in inset) recorded in North America (light blue triangles in inset) on the West-East profile after correction to a common station MLAC relative to the GyPSuM model[5] at a depth of 0−2000 km. Results are plotted at their turning points. Note the sudden change of travel time residuals across the inferred northern boundary of the Pacific Large Low-Velocity Province (LLVP) (purple dashed line). The grey and red dashed lines are LLVP boundaries from He and Wen[24] and Frost and Rost[35], respectively. **b** S travel time residuals on the West-East profile (BB' and CC') in **a** projected on the North-South profile (AA') as color-coded circles with size proportional to residual magnitude at their turning depth. The depth cross-sections are from the GyPSuM model. BB' and CC' profiles are consistent with S wave traveling outside (faster) and inside (delayed) of the LLVP, respectively. The purple line outlines our inferred LLVP surface by separating delayed (magenta) and faster (black) S travel time residuals.

turning depth, it suggests a subvertical edge tilting to the north, which is different from a north-west dipping edge inferred from P-wave study[35]. With limited measurements, we expect to see differences among the obtained dip directions of the LLVP edge. Nevertheless, a north tilting edge better explains the gradual decrease of $\delta t_S$ at larger distance for events A–D along the N-S profile (Fig. 3c). In contrast, an LLVP with a south tilting edge and a wider base produces stronger delays at large distance than observed in the data, in particular for event A at distance larger than 90° (Supplementary Fig. 5).

**North Pacific (NP) ULVZ**. A model that only includes the new LLVP structure as in Fig. 3a cannot reproduce $\delta t_{ScS-S}$ of event B along the N-S profile (Fig. 5a, Supplementary Fig. 6a) and waveforms with late and strong ScS arrivals at a distance range of 90°–100° (Supplementary Fig. 6b), which suggests the existence of a ULVZ at the northern edge. We use the normalized cross-correlation coefficient (CC) between data and synthetics in a time window of 20 s before S and 20 s after ScS (Supplementary Fig. 7) to quantify the goodness of waveform fit (see Methods and Supplementary Note 1). Through the three-dimensional (3D) ULVZ structure, a two-dimensional (2D) slice spanning 12° in the N-S direction with 50±20 km height and −10±2% $\delta V_S$ predicts S and ScS data for all events A–D along this 2D corridor (Supplementary Figs. 8, 9). For this slice, we made series of sensitivity tests on the height, $\delta V_S$, and extent of the ULVZ. In Supplementary Fig. 10, we show that the location and extent of the ULVZ are tightly constrained by matching the waveforms of S and ScS across all distances for the 2D waveform modeling. However, the strong azimuthal variation of $\delta t_{ScS-S}$ (Supplementary Fig. 6) of event B suggests the existence of a 3D ULVZ structure.

The 3D HM ULVZ model[25] near Hawaii is mainly derived from matching the azimuthal variation of strong delayed (>30 s) Sdiff postcursors at the period of 10–20 s along the W-E profile for event F (Fig. 5b), which are refracted from the circular boundary of the ULVZ due to its large velocity reduction. However, for event B along the N-S profile, the HM ULVZ model

produces late ScS arrivals only at an azimuth of 10°–18° (Supplementary Fig. 6b). In contrast, distinct ScS arrivals emerge at an azimuth of ~4° and fade away at ~25° in the data (Supplementary Fig. 6b), which indicates a broader ULVZ extending further to the west (Fig. 1b). The HM ULVZ model also produces strong postcursors after both ScS and sScS (red dashed line in Supplementary Fig. 6a and yellow patches in Supplementary Fig. 6b), which are generated from multiple reflections in the ULVZ due to its strong velocity perturbation. Furthermore, the HM ULVZ model predicts more delayed ScS than data (Supplementary Fig. 6a). Thus, a broader ULVZ in the W-E direction with a weaker velocity perturbation than the HM ULVZ model is required as demonstrated in our 2D modeling.

Our new ULVZ model (referred to as the North Pacific (NP) ULVZ hereafter) fits the Sdiff postcursors along the W-E profile at different frequencies. At the period of 10–20 and 10–50 s, strong Sdiff postcursors extend to the azimuth of 48°, which is predicted by the NP ULVZ (Fig. 5b, Supplementary Fig. 11). The Sdiff at the period of 10–50 s exhibits two arrivals at the azimuth of 48° to 55° and broadening waveforms at larger azimuth (Supplementary Fig. 11). Both the HM ULVZ model and the NP ULVZ predict Sdiff waveforms broadening at azimuth larger than 55°, but neither produces Sdiff with double arrivals at smaller azimuth, which may originate from multi-pathing at the edge of the LLVP.

**Sensitivity tests on the ULVZ model parameters**. The NP ULVZ in Fig. 1b inherits the shape of the HM ULVZ model in the east but extends further to the west. The shear velocity drop of the ULVZ should not exceed 10% to avoid strong arrivals after ScS for event B (Supplementary Fig. 6). The eastern boundary of the ULVZ can be defined at which S and ScS merge along the azimuth profile at an azimuth of ~25° (Fig. 1b, Supplementary Fig. 6b), which is also supported by ULVZ mapping using ScS pre/post cursors[30]. To further investigate how different 3D ULVZ model parameters, i.e. $\delta V_S$, height ($H$), lateral size, shape, density, and possible existence of heterogeneities, affect the waveform complexities for both N-S and W-E profiles, we perform

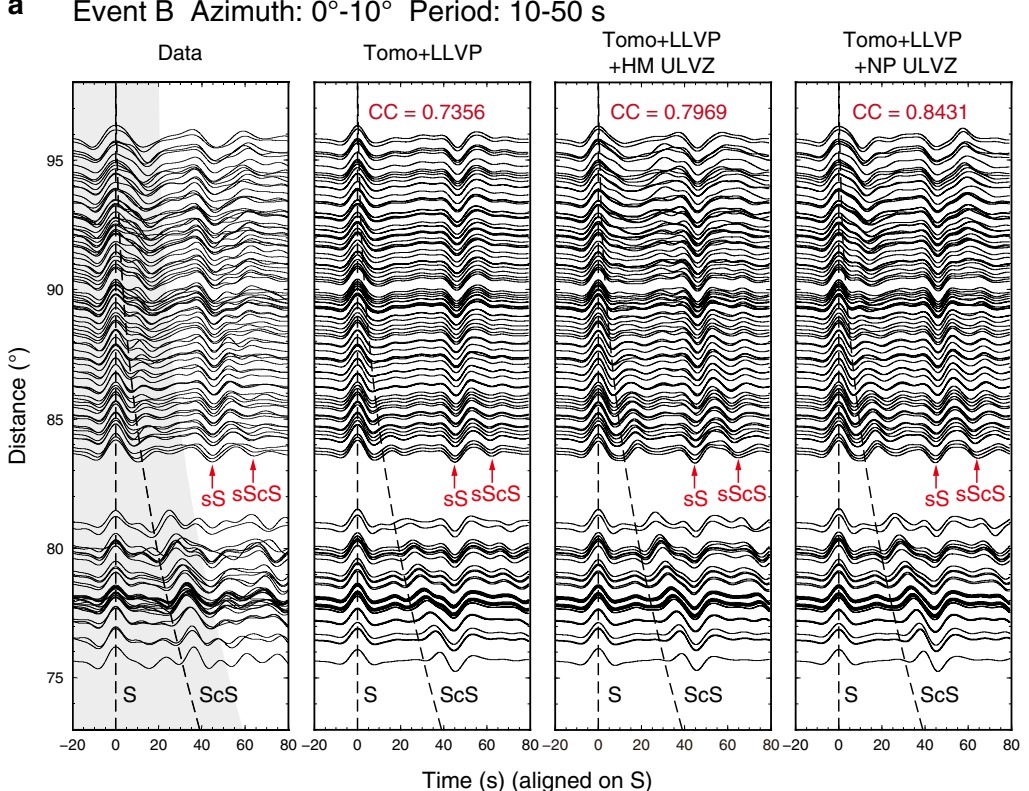

**a** Event B  Azimuth: 0°–10°  Period: 10–50 s

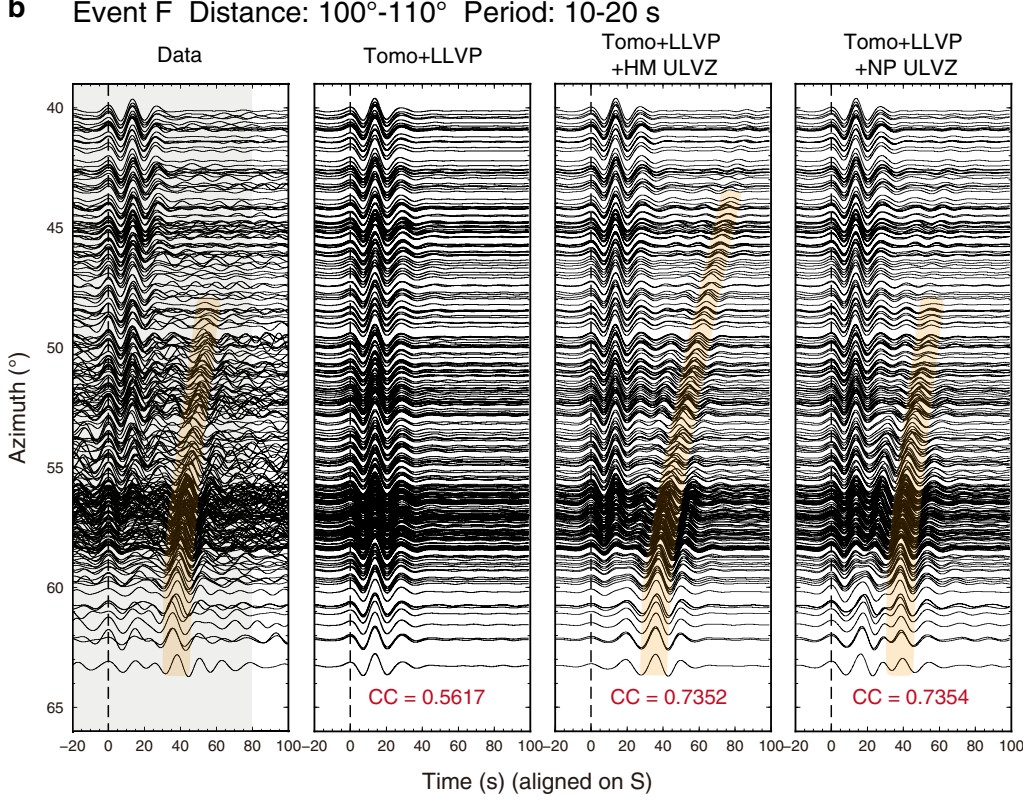

**b** Event F  Distance: 100°–110°  Period: 10–20 s

sensitivity tests (See Supplementary Note 1, Supplementary Figs. 12–15). Based on grid searching, a model with $\delta V_S$ of –10% and $H$ of 50 km provides the highest CC. In Supplementary Table 3, we describe how these model parameters affect waveform

complexities in detail. In addition to using CC as the misfit function, we also test L1-norm and L2-norm (see Methods), which are typically applied in full waveform inversion, to examine the trade-offs between $\delta V_S$ and $H$ of the ULVZ. We find that

**Fig. 5 Comparison between the data and 3D synthetics. a** Distance profiles of event B in 0°−10° azimuth range. Tangential components are aligned on S and bandpass filtered from 10−50 s. The dashed lines in each panel show the S and ScS arrival times predicted by the IASP91 model[34]. The depth phases sS and sScS are also labeled. **b** Azimuthal profiles of event F in 100°−110° distance range. Tangential components are bandpass filtered from 10−20 s. The yellow patches denote the strong postcursors after Sdiff arrivals. Columns in **a** and **b** from left to right are for data, our inferred Large Low-Velocity Province (LLVP) model, LLVP model plus Hawaiian mega-sized (HM) ultra-low velocity zone (ULVZ)[25], LLVP model plus North Pacific (NP) ULVZ, respectively. Cross-correlation coefficients (CC) between data and synthetics are calculated in the time window indicated by the grey shaded region (see Methods for details). The average CC are present in the synthetic panels.

these two misfit functions give similar results as CC (Supplementary Figs. 12–14), implying the CC values used in this study are robust. In general, only modelling Sdiff postcursors from the W-E profile cannot constrain $\delta V_S$ and the shape of the NP ULVZ well. A ULVZ extending along the W-E direction with different $\delta V_S$ can produce Sdiff postcursors similar to those predicted by the circular HM ULVZ model. However, combined with constraints from the N-S profile, we can better define the shape and $\delta V_S$ of the NP ULVZ.

In most cases, we assume a ULVZ model with uniform $\delta V_S$ and height. However, the NP ULVZ could be inhomogeneous considering its large scale. Thus, we test models with heterogeneities as well as multiple separated smaller ULVZs (Supplementary Fig. 15N–Z). Models with isolated ULVZs fail to explain the azimuthal variation on the N-S profile and the Sdiff postcursors on the W-E profile. ULVZ models with a gradual radial or randomly distributed $\delta V_S$ (Supplementary Fig. 15S, T) predict almost identical waveforms as the NP ULVZ model with uniform $\delta V_S$. We also generate pseudo-tomographic models by enhancing global tomographic models or fitting the measured $\delta t_{ScS-S}$ along the N-S profile (see Supplementary Note 1 and Supplementary Fig. 15U–Y). Results for these models suggest that a heterogeneous ULVZ cannot be constrained by fitting the waveform data at the frequency bands in this study. However, the $\delta V_S$ in the NP ULVZ is not likely to vary drastically at long wavelength. The $\delta V_S$ variation should not exceed 20% radially for layered ULVZ models, and not exceed 4% at the wavelength of ~200 km laterally, compatible with the Fresnel Zone of Sdiff at the period of 10 s (Supplementary Fig. 15A). Also, the NP ULVZ must be a continuous block at large scale, rather than many small disconnected patches (Supplementary Fig. 15Z). Despite the possible existence of small-scale heterogeneities, our uniform NP ULVZ model represents the bulk properties of the ULVZ located at the northern boundary of the Pacific LLVP. In addition, CMB topography of 20 km extending over 15°, which is comparable to the size of the NP ULVZ, has negligible influence on the waveform complexities at the frequency bands we focus on (Supplementary Fig. 16). In summary, our preferred 3D ULVZ model has a height of 50 km, lateral extent of ~1500 × 900 km at the CMB, and $\delta V_S$ of −10%, which best reproduces the waveform complexities from the two orthogonal profiles.

**P-velocity reduction of the NP ULVZ.** Luo et al.[28] define a ULVZ (Fig. 1b) overlapping with the NP ULVZ with a P-velocity reduction of ~10% by modeling the late PKPab arrivals and associated waveform complexity. We add more measurements of differential travel time residuals between PKPab and PKPdf ($\delta t_{PKP(ab-df)}$) for events along the Tonga-Fiji-Kermadec region (Supplementary Table 2) recorded by stations in Europe and find that PKPab only displays ~1 s travel time variation and has no obvious amplitude reduction or waveform complexity (Supplementary Fig. 17b). We suspect that the observed waveform complexities and delayed PKPab in Luo et al.'s study may partly originate from strong upper mantle structure. With a $\delta V_P$ of −10%, the NP ULVZ model predicts the $\delta t_{PKP(ab-df)}$ of up to 3 s

(Supplementary Fig. 17c). To match the observed ~1 s variation of $\delta t_{PKP(ab-df)}$ and simple waveforms of PKPab (Supplementary Fig. 17d), a P-velocity drop less than 5% is sufficient to keep PKPab normal compared to data, consistent with a ULVZ with a weak $\delta V_P$.

**Discussion**

The NP ULVZ (Fig. 6a) is about twice the size in both height and lateral extent of the mega-sized ULVZs beneath Iceland[31] and Marquesas[36]. Heterogeneity probability maps of highly anomalous SPdKS waveforms suggest widely distributed mega-scale ULVZs[32,33] along the Pacific Rim beneath Samoa, east of the Philippines and north of Papua New Guinea, and beneath North America and South America, which are comparable in size to the NP ULVZ. Despite the large size of the NP ULVZ, its aspect ratio (relief/width) is similar to the mega-sized ULVZs beneath Iceland and Marquesas.

ULVZs may represent patches of partial melt[14] or iron-rich oxide[13,32], which are denser than the ambient mantle. Compared with the large shear velocity drop of 20–45% for other mega-sized ULVZs[31,36,37], the NP ULVZ has a weaker $\delta V_S$ of −10%, suggesting a less dense ULVZ[13] and possibly different chemistry from other ULVZs. A smaller density tends to form a ULVZ structure with high aspect ratio (relief/width)[38]. Furthermore, a lighter ULVZ tends to be stirred more into the LLVP[39], so replenishment of the NP ULVZ may be necessary to sustain its large size. For the "mild" NP ULVZ with a small velocity drop, its viscosity might be larger than ULVZs with extreme velocity drops[40], which also favors a higher aspect ratio. Thus, the similar aspect ratio of the NP ULVZ and other mega-sized ULVZs may indicate the chemistry or dynamics of the ULVZs are different. The location of the NP ULVZ at the northern edge of the Pacific LLVP, the large size of the ULVZ, and $\delta V_S/\delta V_P$ of ~2 suggests the ULVZ has distinct chemistry rather than solely partial melt, the latter of which favors a $\delta V_S/\delta V_P$ of ~3[14,41]. However, without precise constraints on the elastic and thermal properties of the ULVZ, it is difficult to distinguish the origin of the ULVZ. Iron-rich assemblages[13,42], subducted related materials[15,43–45], and partial melting may contribute to the chemical heterogeneity of the ULVZ. Due to a strong overlying hot upwelling, which has been imaged as "broad plumes"[46], the mega-sized ULVZs beneath Iceland[31] and Samoa[32,33] may be more elevated and therefore have a higher aspect ratio than expected. Such a strong hot upwelling could also entrain and drain the ULVZ materials more efficiently than the NP ULVZ[38,47], which further supports a more massive NP ULVZ compared to Iceland and Samoa.

Here we hypothesize that a long-term, stable, horizontally convergent mantle flow plays an important role in driving the migration of "mild" ULVZs in the north Pacific towards the northern wall of the Pacific LLVP and producing the NP ULVZ (Fig. 6c). Numerical models show flow in the deep lower mantle under the North Pacific from 100–50 Ma due to long-lasting subduction history[48–51]. Moreover, the Pacific LLVP could have been stable prior to Pangea at 330 Ma[17]. Thus, the long-lived Pacific LLVP and stable convergent mantle flow towards the northern edge of the

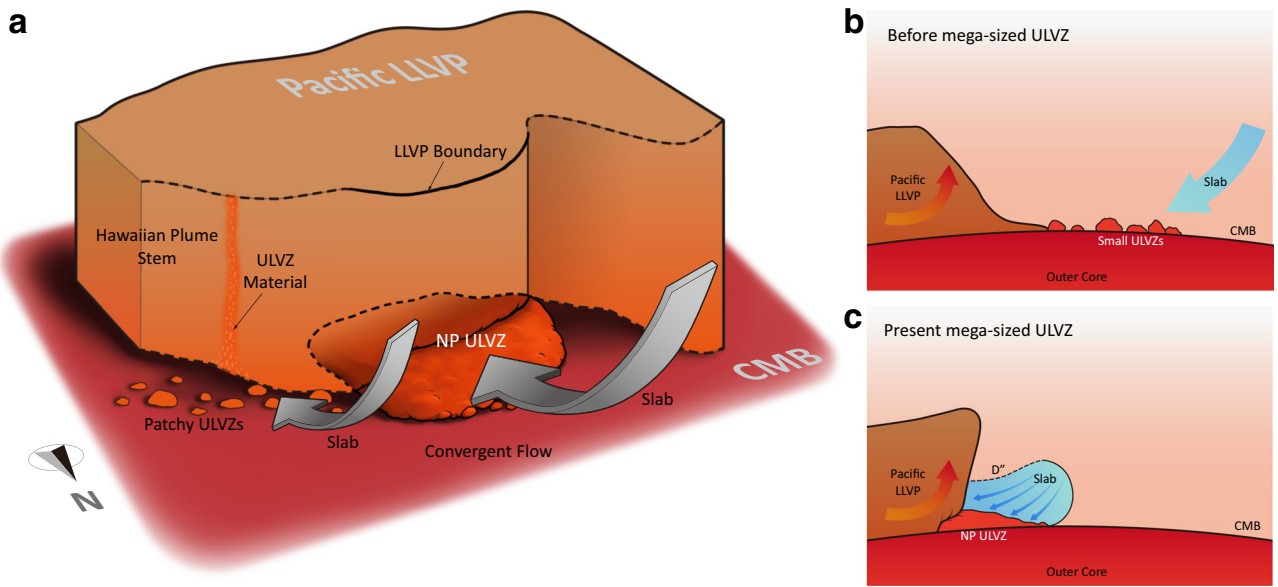

**Fig. 6 Evolution of the ultra-low velocity zone at the northern edge of the Pacific Large Low-Velocity Province. a** Locations of ultra-low velocity zones (ULVZs), Large Low-Velocity Province (LLVP) boundary and the deep-rooted plume near Hawaii, which is rooted at the northeastern edge of the Pacific LLVP with a north tilting plume conduit. **b** Small ULVZs are driven by stable convergent mantle flow towards the LLVP margin with slabs presenting near the edge of the LLVP. **c** These ULVZs are continuously piled up at the LLVP edge and aggregate to form the North Pacific (NP) ULVZ. Also, the convergent mantle flow develops an LLVP edge with high relief that tilts to the north.

Pacific LLVP may both promote and sustain the accumulation of ULVZ material at the LLVP edge[39,52] (Fig. 6b).

The stable convergent mantle flow in the North Pacific is also evidenced by the northward tilting wall of the Pacific LLVP. The LLVP has a subvertical northern wall (Fig. 4b) which distinguishes it from a dynamic ridge-like structure with gradual sloping boundaries[10]. Such tilting away from the center of the LLVP is also observed along the eastern boundary of the African LLVP due to the mantle flow associated with the motion of the African plate[9,53]. A D″ discontinuity, characterized by 2% $V_S$ jump at 220 km above the CMB[24], most likely linked to subducted slab, is also identified at the northern edge of the Pacific LLVP. Thus, the base of the Pacific LLVP may be impinged by slabs and associated stable convergent flow to develop a subvertical tilting wall (Fig. 6c). Walls that tilt away from the center of an LLVP indicate that LLVP materials cannot be too dense, otherwise ridges form with inward tilting walls. Also, LLVP materials cannot be too light, otherwise they produce unstable structures that do not persist through geological time[9,48].

Interestingly, at the northeastern edge of the Pacific LLVP only patchy ULVZs are observed[27,29,54,55]. Hence based on our hypothesis of forming large-scale ULVZs, one may expect stable convergent mantle flow is absent in this region[56] or these patchy ULVZs are fragments from a large-scale ULVZ. The later hypothesis is supported by dynamic calculations that show part of a large ULVZ advected into a linear shape along the LLVP boundary and fragmented into small patches by shearing flow[39]. Subsequently, these fragmented ULVZs may stir into the LLVP[39] and entrain into the Hawaiian plume due to slab push[29], supplying enriched geochemical heterogeneities in the plume. In any case, in a region with a large volume of pre-existing small ULVZs, a strong and stable convergent mantle flow may promote the formation of a mega-sized ULVZ such as the NP ULVZ. Thus, in future work, resolving detailed deep mantle flow patterns will be critical to deciphering both the geographic and geometric evolution of ULVZs. In turn, a detailed global map of ULVZs will provide extra constraints on the direction of deep mantle flow.

Flow in the deep mantle could produce a southward motion of the north tilting Hawaiian plume[57], such that the current root of the Hawaiian plume is likely located south of the Hawaiian Islands[49]. Despite challenges in resolving the Hawaiian plume[58], tomography models (depth cross-sections in Supplementary Fig. 18) and thinning transition zone identified beneath the south-southeast of Hawaii[59,60] suggest the Hawaiian plume is rooted at the northeastern edge of the Pacific LLVP rather than the NP ULVZ. Thus, patchy ULVZs at the northeastern edge of the Pacific LLVP are likely be a plume generation zone[57]. In contrast, a recent study focusing on ScS pre/post-cursors further confirms the existence of a large-scale ULVZ beneath the west of Hawaii and prefers a closer relationship between the NP ULVZ and the Hawaii plume[30]. Nevertheless, to further understand how deep reservoirs fuel near-surface melting and volcanism, constraints on the physical properties of ULVZs and their spatial relation with the LLVPs and slabs are required. Thus, a detailed global mapping of the multi-dimensional structure of ULVZs using sampling from different directions will be crucial.

## Methods

**Data processing procedures and travel time measurements**. We select 13 events with simple source time functions between 2016 and 2020 along the Tonga-Fiji-Kermadec region recorded by USArray stations in Alaska (Fig. 1, Supplementary Table 1) to examine the S-wave velocity structure at the northern edge of the Pacific LLVP. Raw data have been deconvolved with their instrument responses and bandpass filtered from 5–50 s before rotation into tangential (SH) and radial (SV) components. We also include 6 events (Supplementary Table 1) in the Solomon Islands recorded in North America to provide orthogonal direction sampling for an extra constraint on LLVP structure.

For the 13 events (Fig. 1) in the Tonga-Fiji-Kermadec region recorded by stations in Alaska, we measure the ScS-S differential travel time residuals ($\delta t_{ScS-S}$) with epicentral distances smaller than 85°, at which S and ScS are well separated. For event B, we extend the epicentral distance to 90° with the high-quality data of ScS. Then we correct the travel time residuals to one common station (AK.UNV) to remove possible errors of event origin time and location. For the W-E profile with events in the Solomon Islands recorded by stations at North America, we shift $\delta t_S$ to a common station MLAC to remove possible errors of the event origin time and location (Fig. 4a, Supplementary Fig. 4).

The ray paths of PKPab and PKPdf separate at deeper depth with PKPab piercing points covering a wider range at the CMB on the source side than on the receiver side and vice versa for PKPdf (Supplementary Fig. 17). Since small seismic velocity changes on the CMB can perturb the PKPab path effectively, ULVZs can be detected by $\delta t_{PKP(ab-df)}$. Here, 9 events in the Tonga-Fiji-Kermadec region (Supplementary Table 2) are selected for PKP study. All PKP data are selected with epicentral distance at which the PKPab and PKPdf are well separated. Travel time differences of each trace between PKPab and PKPdf are measured by cross-correlating the waveforms of PKPab and Hilbert transform of PKPdf when the amplitudes are normalized to PKPdf. The cross-correlation coefficients of all traces of each event are then normalized to the highest coefficient of that event.

**S-wave travel time correction of events A–D.** The stations at Alaska are located above the Aleutian subduction system, which could include lateral varying structures. Thus, a correction of the travel time anomaly caused by shallow structures beneath the stations is necessary to determine the structure of the LLVP accurately. Here, we test both SKS and SS as reference phases to correct S-wave travel time. Note that the incident angles of SKS and SS are steeper and shallower than that of S, respectively (Supplementary Fig. 2a).

We use SKS travel time residuals from event E located further to the south to correct S-wave travel time residuals for events A–D. SKS ray paths are away from the Pacific LLVP (Supplementary Fig. 2a), so the SKS travel time residuals avoid being contaminated by the Pacific LLVP anomaly. Furthermore, SKS signals are simple in the distance range for event E to provide stable measurements. Supplementary Fig. 2b displays travel time residuals of SKS (red circles) up to ~3 s with increasing distance, which correlates with a slow mantle wedge beneath Alaska. For comparison, we also measure SS travel time residuals of events A–D whose ray paths are almost identical in the uppermost mantle beneath the source and station. Supplementary Fig. 2b displays travel time delays of SS (green circles) of event B as an example. Despite larger uncertainties in measuring the travel time of SS, the travel time residuals of SS are similar to those of SKS.

Supplementary Fig. 2c shows the corrected S travel time residuals by using both SKS from event E and SS from individual events A–D, which have similar trends. Such consistency implies that the observed travel time anomalies of SKS and SS mainly come from the crust and upper mantle anomaly beneath Alaska and the source side structures of events A–D have little effect. Thus, the lower mantle LLVP, which is only sampled by the S, contributes most of the travel time change of S after correcting SKS or SS travel time anomalies.

The height of the Pacific LLVP determines the initial epicentral distance of the $\delta t_S$ jump, at which the corresponding S wave starts to sample the top of the LLVP. Supplementary Fig. 2c shows that the initial epicentral distance where $\delta t_S$ increases is identical for different corrections. Therefore, whether the SKS or SS correction is applied, the height of the Pacific LLVP is a robust result. We do notice that $\delta t_{SKS}$ are slightly larger than $\delta t_{SS}$ by ~0.5–1 s. Thus, depending on which reference phase is chosen, the absolute value of the $\delta V_S$ of the LLVP can vary. For example, compared to SKS as the reference phase, if we use SS as the reference, the velocity perturbation is smaller (~1% less reduction of $\delta V_S$ assuming the same geometry of LLVP).

**Synthetic seismograms.** For most cases, we perform 3D wave propagation calculations using the SPECFEM3D GLOBAL package[61] with the shortest period at ~8 s. Thus, we focus on modeling the data at the period of 10–50 s. For the N-S profile, we also apply a 2D finite-difference code, which has been widely used to generate 2D global synthetics at high frequency with high efficiency[29]. The grid spacing in the 2D run is 2 km, which allows accurate simulation up to 3 Hz. Hence for 2D simulations we generally bandpass filtered data from 5–50 s. We apply the same processing procedures as for the data. The focal mechanisms used in the simulations are selected from the USGS earthquake catalogue (http://earthquake.usgs.gov) and the Global Centroid-Moment-Tensor (CMT) solutions (https://www.globalcmt.org).

**Misfit between data and synthetics.** The definition of the misfit function, the misfit between data (D) and synthetics (S), quantifies how well we model the data. Here, we first calculate the average normalized cross-correlation coefficient (CC) to evaluate the goodness of waveform fit between data and synthetics as:

$$CC = \frac{1}{M}\sum_j^M \left( \frac{1}{N}\sum_i^N CC_i(D, S) \right) \tag{1}$$

where N is the number of traces for each event and M is the number of events.

For the N-S profile, the CCs are calculated within a time window of 20 s before S and 20 s after the ScS predicted by IASP91 (Supplementary Fig. 7). For the W-E profile, the time window is fixed as 20 s before and 80 s after Sdiff to include the postcursors as shown in the shaded region in Fig. 5.

In full-waveform inversions, a typical approach is to use the L2 norm of waveform difference between data and synthetics as the misfit function. Thus, we also use L1-norm and L2-norm as misfit function to examine trade-offs between $\delta V_S$ and H in the 3D simulations (Supplementary Figs. 13, 14). For an individual event, the L1-norm misfit is defined as

$$\sigma_{L1} = \frac{1}{N}\sum_i^N \int_{t0}^{t1} |D_i(t) - S_i(t)|dt \tag{2}$$

and the L2-norm misfit is defined as

$$\sigma_{L2} = \frac{1}{N}\sqrt{\sum_i^N \int_{t0}^{t1} (D_i(t) - S_i(t))^2 dt} \tag{3}$$

where N is the number of traces. Here, both $D_i$ and $S_i$ are aligned and normalized to the S and Sdiff for the N-S and W-E profile, respectively. The time window selections of $[t_0, t_1]$ in the L1-norm and L2-norm calculations are the same as that in the calculation of CC.

## Data availability

All waveform data are archived and openly available at the IRIS Data Management Center (www.iris.edu/ds), under station code TA for the USArray Transportable Array (IRIS Transportable Array, 2003; https://doi.org/10.7914/SN/TA), and SCSN (Southern California Seismic Network, 1926; https://doi.org/10.7914/SN/CI).

## Code availability

The open source SPECFEM3D GLOBAL package used in this study is available at https://github.com/geodynamics/specfem3d_globe.

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

## Acknowledgements

We thank the IRIS Data Center and Earthscope USArray for providing the seismic data. All maps in this paper were produced using GMT developed by Paul Wessel and Walter H.F. Smith. We are thankful to the B-type Strategic Priority Program of the Chinese Academy of Sciences, Grant No. XDB41000000, XDB18000000, National Natural Science Foundation of China 41722401 for support of this research. D.J.B acknowledges Swiss National Science Foundation (SNSF) Ambizione Grant 173992. We appreciate the Supercomputing Center of USTC, the Tianhe platforms at the National Super-computer Centers in Tianjin and Thanhe2 platform at Guangzhou for high-performance computing services. We also thank Lei Li for helping with the artistic illustration of Fig. 6.

## Author contributions

All authors contributed to the manuscript. D.S. designed the project. J.L. performed the seismic studies. D.J.B. provided discussion and interpretation. J.L. wrote the first draft of the paper. D.S. and D.J.B. commented and contributed text to the manuscript.

## Competing interests

The authors declare no competing interests.
