## [Peer Review File · Nature Communications]

Reviewers' comments:

Reviewer #1 (Remarks to the Author):

This paper examines a variety of seismic wave data, primarily focusing on S and ScS wave arrivals, to infer a low-velocity zone north of the Pacific Low Velocity Province. It is stated that this is the largest ultra-low velocity zone yet detected and that it is likely related to convergence at the LLSVP edges. The work done by the authors of this paper is very extensive, especially in regards to the synthetic model predictions and in constructing a model that appears to match these exceptionally interesting data. However, this paper is a real challenge to read in its present state. This paper jumps from one observation to the next without preamble and often doesn't describe what it is actually referring to. There are many instances in reviewing this paper that I really struggled to figure out what the authors were talking about. In the end the authors suggest a rather large ULVZ model, that they refer to as the NP ULVZ. But, it is a challenge in reading this paper to assess this model as nothing quantitative is really produced. Waveforms are presented, which are certainly intriguing and look like they may do a good job of predicting these data, but we are mostly left with statements to the effect of we did this so that we didn't get big pre-cursors. Thus, I found it difficult to assess whether or not the modeled ULVZ is really adequate. Also, the primary conclusions I find are either (a) not supported in that to state this is the largest ULVZ discovered to date by far, as the authors state, disregards recent studies and thus is not correct, or is (B) not necessarily novel, in that several previous studies have already made suggestions of ULVZs accumulating near the boundaries of LLVPs. However, I think this observation of a potentially very large ULVZ in this location is really interesting and ultimately, I would love to see this paper get published as it will be of great interest to a large audience. However I think that it requires major effort in reorganization and clarification before it will be ready for publication. Below I include a listing of some of the challenges I had in reading this manuscript.

1. The largest ULVZ. The authors make many claims in this paper that the ULVZ discussed in this paper is the largest ULVZ ever discovered. This statement is included in the title. "Fragmentation of the largest Ultra-low Velocity Zone beneath the Pacific". Again this is stated on line 198, "The NP ULVZ is by far the largest imaged ULVZ, which about twice the size in both height and lateral extent of the mega-sized ULVZs beneath Iceland, Samoa, and Marquesas." The claim in this paper is that this ULVZ has height of 50 km, and lateral extent of ~1500 x 900 km. Based on this measurement, this particular ULVZ may likely be characterized as among the largest ULVZ ever discovered. But, this ignores recent work showing that many mega-sized ULVZs may exist of comparable size (see e.g., Thorne et al. 2020 and 20201). This particular ULVZ seems relatively on par in size with the Samoa ULVZ or may even be a bit smaller (which is reported to be on the order of 1600 x 500 km). Hence, I suggest the authors remove statements that appear to me to be incorrect that this NP ULVZ is among the largest.
2. Line 52: the authors declare that 5-15% P-wave reductions are typical of ULVZs but larger P-wave velocity reductions on the order of 20-25% are certainly being reported see, e.g., Brown et al. 2015.
3. Line 58: "...likely the ULVZs are likely to distribute over LLSVP margins...". This is somewhat unclear to me that this is the case. Yes, if you only use ScP and throw out all other ULVZ observations as reported in Yu and Garnero, 2018. However, if you include SPdKS data then it seems that ULVZs are not necessarily tied to LLSVP margins rather occur in all lower mantle settings. See, Thorne et al. 2021.
4. Also, LLSVP is maybe better stated as LLVP, since recent P-wave velocity models are also seeing large low velocity provinces (see e.g., Hosseini et al., 2019 or Simmons et al. 2010).
5. Line 64: the authors state that constraints from seismic rays are required. I would change this to seismic waves. Seismic rays are an artificial construct.
6. Lines 66-68: "However, defining the LLSVP beneath the Pacific is challenging because of limited data coverage. Thus, most studies focus on characterizing the edge of the Pacific LLSVP, and in particular ULVZs at its edge." I really don't understand what you are saying here. It sounds like you are saying it is challenging to investigate the interior of the Pacific ULVZ and thus most studies focus on the edges of the Pacific ULVZ where data coverage is better. Thus, (and in addition), most studies of ULVZs in relation to the Pacific ULVZ are also concentrated near the edge. Perhaps the authors could improve the readability of this part.
7. Lines 73-79. No mention of mega-sized ULVZs even larger than Iceland or Hawaii, again ignores other potentially larger ULVZs than just Iceland and Hawaii.

8. Line 88: What are you calling δt_{ScS} and δt_S ? I assume you mean the difference of the ScS and direct S-wave arrival respectively with respect to some reference model such as PREM. But you don't actually tell us? I see that actually you make a statement in one of the figure captions that this may be referenced against IASPEI. Please clarify in the text.

9. Line 97: "For the northernmost event D, the change of dts is only up to 5 s, suggesting a contribution from the shallow structure beneath Alaska." I have no idea what this means. There is a 5 s change from event A also, and a 6 s change from the next event. Why does this particular event suggest shallow structure and the others do not?

10. Line 102: Here the authors state that they use the travel-time anomaly of SKS for event E to correct for upper mantle structure. This is completely opaque to as to why. The SKS path has almost no overlap with any of the modeled events, except perhaps for a small region immediately beneath Alaska. What thickness range beneath Alaska does this correct for? What do you do about the anomalous source-side of structure that SKS for this event encounters on its way down? I feel that this needs to be further justified, as to why this SKS path is representative of the upper mantle beneath Alaska and not just contaminated by other structure along the way.

11. Line 108: "the onset distance of the dts jump". It is not clear what this means. What dts jump are you referring to? Does this refer to the transition (such as shown in Fig. 2d) between positive and negative dts values.

12. Line 108: continuing the above point, the authors state this is where "...S is starting to sample the LLSVP which puts a good constraint on the height of the Pacific LLSVP." It appears first of all that the largest anomalies is where the S-wave is faster (i.e., black open circles in Fig. 2d). Here S is advanced by as much as 2 s, whereas it appears to only be delayed by 1 s in the LLSVP region. In this case it seems that there is some fast anomaly that dominates the signal, and not necessarily the LLSVP. Also, I don't see how much constraint is given on the height of the LLSVP through these data? The next sentence makes a statement trying to clarify this point, but perhaps drawing some ray paths would help to see how much height difference there is.

13. Line 111: Now the authors immediately jump into their "preferred LLSVP model" but no information is given on how they arrive at this preferred model.

14. Starting at line 136: The authors jump in to stating a hybrid LLSVP model and a mega-sized ULVZ model are inadequate in explaining ScS data, but never do they really define what these models are. The hybrid LLSVP model is presumably the "preferred" model they constructed somewhere at some time, possibly in this paper. But what is the mega-sized ULVZ model they refer to?

15. Line 149: "A ULVZ model ... With 50 km height and -10% dVs predicts S and ScS data for all events...". OK, but is there not a fundamental tradeoff between dVs and thickness for S and ScS such that a thinner ULVZ with strong dVs might also work? The uncertainties and tradeoffs need to be discussed. It's impossible to tell from extended data figure 6 that another model doesn't work.

16. Fig 2. In panel b the y-axis goes up to 20 s. Perhaps this could be improved if it was cut down to 10 s. The observations aren't larger than 10 s, so it might be easier to see the variation in the dt measurements if the y-axis scale is reduced.

17. Fig 3. In panel b, the synthetics actually look like they capture the nature of the data well. However, can something more quantitative also be shown, for example how much does the timing of the post-cursor differ for data and the (Tomo+LLSVP+mega-sized ULVZ) relative to the (Tomo+LLSVP+NP ULVZ) model? That is, I can't visually inspect these seismograms and say that oh yes, one model definitely performs better than another.

References

Brown, S.P., Thorne, M.S., Miyagi, L., and Rost, S. (2015), A compositional origin to ultralow-velocity zones, *Geophysical Research Letters*, 42, 1-7, doi:10.1002/2014GL062097.

Hosseini, K., Sigloch, K., Tsekhmistrenko, M., Zaheri, A., Nissen-Meyer, T., & Igel, H. (2019). Global mantle structure from multifrequency tomography using P PP and P-diffracted waves. *Geophysical Journal International*, 220(1), 96-141. <https://doi.org/10.1093/gji/ggz394>

Simmons, N. A., Forte, A. M., Boschi, L., & Grand, S. P. (2010). GyPSuM: A joint tomographic model of mantle density and seismic wavespeeds. *Journal of Geophysical Research*, 115(B12). <https://doi.org/10.1029/2010jb007631>

Thorne, M.S., Pachhai, S., Leng, K., Wicks, J.K., and Nissen-Meyer, T. (2020) New candidate ultralow-velocity zone locations from highly anomalous SPdKS waveforms, *Minerals*, 10 , 1-26, doi: 10.3390/min10030211.

Thorne, M. S., Leng, K., Pachhai, S., Rost, S., Wicks, J., & Nissen-Meyer, T. (2021). The most parsimonious ultralow-velocity zone distribution from highly anomalous SPdKS waveforms. *Geochemistry, Geophysics, Geosystems*, 22, e2020GC009467. <https://doi.org/10.1029/2020GC009467>

Reviewer #2 (Remarks to the Author):

The manuscript by Li et al presents a nice study of ultra low velocity zone (ULVZ) at the northern edge of the Pacific LLSVP using two orthogonal seismic ray paths. They performed ScS-S differential travel time analysis using dense array data in Alaska and North America for earthquakes in Tonga-Fiji-Kermadec. They further adopted both SPECFEM3D global package and a 2D finite difference code to compute synthetic waveforms, which can explain major features of observations. Then, they proposed a dynamic picture for origin of such large ULVZ.

Their analysis and results are generally organized. The novelty of this manuscript is that they declare to identify the largest ULVZ yet detected at the northern edge of the Pacific LLSVP. The paper does contribute an important finding for the deep Earth community in understanding of lowermost mantle structure. However, I do have some concerns about their explanations and some technical questions. I suggest the authors carry out more synthetic tests to support their explanations. The manuscript is recommended to be accepted after a major revision.

General Comments:

In their explanations for the NP ULVZ, they suggest a less dense ULVZ because of a smaller δV_s of -10% compared with other mega-sized ULVZs. However, the authors did not take into account the density perturbations of ULVZ in their simulations. I understand the heavy computations required in a global 3D simulation, but it would be better if the authors could show how the density of ULVZ affect the synthetic waveforms in their sensitivity tests.

The authors constructed the NP ULVZ model as a whole, is it possible that their ULVZ model consists of two individual patches of ULVZs, one is similar as the previously-inferred mega-sized ULVZ, the other one is a smaller-sized ULVZ. The author should exclude this possibility that the NP ULVZ is not composed of separated patches of ULVZs.

I appreciate that the authors also analyzed the additional P-wave data to provide more constraints on the ULVZ properties. According to their PKP results, the dV_p/dV_s of the NP ULVZ is about 1:2. However, the negative 5% of P wave velocity is not in agreement with results of Luo et al. (2001). If the result from Luo et al. (2001) is taken, the dV_p/dV_s ratio of the ULVZ is 1:1, possibly suggesting a compositional origin. Thus I suggest the authors perform more detailed analysis of PKP data that approximately sample the same ULVZ region at the northern boundary of the Pacific LLSVP, because it is important in determining whether ULVZs represent patches of partial melt or iron-rich oxide. Based on this, the authors could shed more light on the origin of ULVZ in this area.

Could the authors use a quantitative measure of fit to determine the best-fitting models? Currently, the authors only compared the observations and synthetics visually. For example, in Fig. 3a, I cannot see much differences in the synthetics between LLSVP model plus mega-sized ULVZ and LLSVP model plus the

preferred NP ULVZ. In contrast, in Fig. 3b, the synthetics from LLSVP plus mega-sized ULVZ model are visually better than those from LLSVP plus NP ULVZ model for the distance range of 50-65°.

Specific Comments:

1. How did the authors choose the source time function in their simulation?
2. It would be better to add the Pearson's r-value in the Extended Data Fig. 1, so that people could know how the results improve after the corrections.
3. Line 91: the δt_{ScS} and δt_{SS} appear constant along the distance profile, I could not get the same statement from Extended Data Fig. 2b.
4. Line 102-104: The authors used δt_{SKS} from event E to correct for upper mantle structure beneath Alaska. According to the raypath of SKS, how would the author rule out the effects from source-side heterogeneity within LLSVP on the δt_{SKS} when the raypaths of SKS pass through?
5. I am a little skeptical about the inferred heights of the authors' preferred LLSVP, if the authors cannot take into the upper mantle heterogeneities properly. The used five events are widely distributed, how did the authors exclude the effects on δt_S from upper mantle heterogeneities in the source regions when considering the new LLSVP model? How did the authors calculate the S wave travel times in 3D models, based on 1D or 3D ray tracing methods? In Fig. 2d and Extended Data Fig. 3, the upper height of the colored circles is less than the top height of the preferred LLSVP (purple line), then how would the authors infer that the LLSVP model has the height of 920 km?
6. The authors use the PKPab and PKPdf data to further constrain the P wave velocity of NP ULVZ. Which frequency band did the authors use to filter the PKP data? The author should show an example of observed PKP seismograms as well as comparison between the observations and synthetic waveforms based on the NP ULVZ model.
7. How would the authors evaluate the effects from the receiver-side structures beneath the Europe when they calculate the PKP differential travel time residuals?
8. The authors measure the differential time residuals relative to PREM for PKP, but for ScS-S measurement, they use iasp91 model as reference. I am wondering why the authors did not use the same model for consistency.
9. The filter bands for S waves are sometimes confusing. In the main text, the filter band is 10-50s, but in the Supplementary Information, the filter band is 5-50s (Line 38), please make them consistent.
10. In Fig. 3, why did the author choose a different filter band (10-20 s) for the event F?
11. Figure 4 b-d look just fine for explaining the accumulation and fragmentation of the NP ULVZ. The authors could have plotted a more beautiful figure.

Line-by-line comments:

1. Line 49: fine-scale structure => fine-scale structures
2. Line 87-88: Please refer to the model through which the authors compute the travel time residuals.
3. In Line 86, the authors state that, the epicentral distances are smaller than 85° for ScS time residuals, however, in Line 90, the distance for δt_{ScS-S} is from 75° to 90°, which is confusing. Please clarify this.

Reviewer #3 (Remarks to the Author):

This paper presents a very nice data set, and a fair amount of modeling. The topic is important. In my opinion, for the paper to be suitable for a high profile journal, something new and novel of impact needs to be presented. This paper attempts making the case that a very large (mega) ulvz is broken into pieces. If that were indeed demonstrated, and a robust demonstration as to what it means, then I agree it is impact science. However, I do not agree that that conclusion is constrained at all. Even if the modeling were better explained (I think I figured out what was being done), the trade-offs are too large. The assumed constant property ulvz is unrealistic, especially to hold that fixed and then model shapes, when the shear velocity drop is relatively weak. If the CMB started with ulvz typical shear drops (e.g., 30%) and then transitioned to smaller amounts, what happens? For something that large, one does not expect constant property. I also struggled with visual side by side comparisons between data and synthetic seismograms. That is quite qualitative. I see many features suggesting fits are not that perfect, which signals trade offs are possible. Only map view structural differences are shown, but that trades off, obviously, with vertical extent and properties, which are discussed faintly. In summary regarding the seismology, I think the data look nice, the modeling looks nice, but I do not see demonstration that the preferred structures are so unique. There is no compelling demonstration that the data are not affected by the source region (since waves leave similar to trench strike, which often results in slab multipathing). These are my first and biggest issues.

My second issue concerns interpretation. Flow model implications are important, but I struggle to believe any particular flow model because all of the assumptions involved. I think that citing dynamics flow behavior from a convection model without connection to any particular geography seems okay for wisdom on general flow behavior (and seems to be done in papers), but tying a particular flow to some specific spot requires a lot of faith. In fact, the same goes with tomography – discussing subvertical walls... those types of features may be very model dependent, depending upon the location. It also depends upon path coverage (both the African and Pac llsvps have tilts along raypaths). One model was chosen, which is referenced for this kind of interpretation work a lot less than Ritsema or French & Romanowicz. There are many papers discussing the variability of models. So I feel the dynamics and reference to tomography details too speculative. A lot of adjectives in description of dynamics were excessive and unfounded/undemonstrated in imaging studies. E.g., line 226 “vigorous” flow, “rapid” southward motion. Later, ‘slab debris deforming the LLSVP wall’ is also pure speculation. There is a lot of that in this paper, and it is pure speculation. The argument for two different territories (line 234) is also unfounded, since the Iceland structure was modeled solely with diffracted waves (thus low resolution).

Thus, while I do appreciate the nice data and hard work of the modeling, I do not think the conclusions and preferred models are constrained.

Dear Editor Sebastian Mueller and reviewers,

Thank you for the opportunity to improve our previously submitted manuscript. We appreciate the clear feedback and helpful comments from you and the three reviewers. We have made significant changes to address all concerns, especially including quantitative assessment of the uniqueness of parameters for the modelled ULVZ, such as density, shape, and trade-offs between velocity and height. We also explore models including heterogeneities and spatially separated ULVZs. Importantly, our sensitivity tests for ULVZ models demonstrate that our NP ULVZ model provides the best overall fit to the ULVZ structure along the northern edge of the Pacific LLVP. Furthermore, we reorganized the main text and supplementary materials as well as figures to enhance the readability of the manuscript. Hence, our manuscript has been greatly improved by all of the comments that we previously received.

The quotes from the reviewers are followed by our responses in *blue italics*.

Review #1 comments

Q1: The largest ULVZ. The authors make many claims in this paper that the ULVZ discussed in this paper is the largest ULVZ ever discovered. This statement is included in the title. “Fragmentation of the largest Ultra-low Velocity Zone beneath the Pacific”. Again this is stated on line 198, “The NP ULVZ is by far the largest imaged ULVZ, which about twice the size in both height and lateral extent of the mega-sized ULVZs beneath Iceland, Samoa, and Marquesas.” The claim in this paper is that this ULVZ has height of 50 km, and lateral extent of ~1500 x 900 km. Based on this measurement, this particular ULVZ may likely be characterized as among the largest ULVZ ever discovered. But, this ignores recent work showing that many mega-sized ULVZs may exist of comparable size (see e.g., Thorne et al. 2020 and 2021). This particular ULVZ seems relatively on par in size with the Samoa ULVZ or may even be a bit smaller (which is reported to be on the order of 1600 x 500 km). Hence, I suggest the authors remove statements that appear to me to be incorrect that this NP ULVZ is among the largest.

A1: Thank you for pointing out Thorne et al.'s series of papers regarding the mega-sized ULVZ. We have included discussion and comparison on the global distributed mega-sized ULVZs at Line 206-Line 210. We also change the title as “Fragmentation of the Enormous North Pacific Ultra-low Velocity Zone”.

Q2: Line 52: the authors declare that 5-15% P-wave reductions are typical of ULVZs but larger P-wave velocity reductions on the order of 20-25% are certainly being reported see, e.g., Brown et al. 2015.

A2: We have changed this sentence to “...characterized by a reduction of 5-25% in compressional and 10-45% in shear velocity” in Line 40-Line 41 and have added the reference as well.

Q3: Line 58: “...the ULVZs are likely to distribute over LLSVP margins...”. This is somewhat unclear to me that this is the case. Yes, if you only use ScP and throw out all other ULVZ observations as reported in Yu and Garnero, 2018. However, if you include SPdKS data then it seems that ULVZs are not necessarily tied to LLSVP margins rather occur in all lower mantle settings. See, Thorne et al. 2021.

A3: Thank you for pointing this out. I do agree that the conclusion of ULVZs distributed over LLVPs margins are biased by the limited observations. Many observations did suggest that ULVZs present at regions other than LLVPs. Thus, we have modified this sentence as “...many ULVZs are likely to distribute over LLVP margins.” in Line 46.

Q4: Also, LLSVP is maybe better stated as LLVP, since recent P-wave velocity models are also seeing large low velocity provinces (see e.g., Hosseini et al., 2019 or Simmons et al. 2010).

A4: We have changed “LLSVP” to “LLVP” throughout the manuscript.

Q5: Line 64: the authors state that constraints from seismic rays are required. I would change this to seismic waves. Seismic rays are an artificial construct.

A5: We have changed “seismic rays” to “seismic waves”.

Q6: Lines 66-68: “However, defining the LLSVP beneath the Pacific is challenging because of limited data coverage. Thus, most studies focus on characterizing the edge of the Pacific LLSVP, and in particular ULVZs at its edge.” I really don’t understand what you are saying here. It sounds like you are saying it is challenging to investigate the interior of the Pacific LLSVP and thus most studies focus on the edges of the Pacific LLSVP where data coverage is better. Thus, (and in addition), most studies of ULVZs in relation to the Pacific LLSVP are also concentrated near the edge. Perhaps the authors could improve the readability of this part.

A6: We have reworded this sentence as: “However, because of limited data coverage, most studies of the Pacific LLVP focus on characterizing its edge structure, and in particular ULVZs” in Line 53-Line 55.

Q7: Lines 73-79. No mention of mega-sized ULVZs even larger than Iceland or Hawaii, again ignores other potentially larger ULVZs than just Iceland and Hawaii.

A7: Here we try to emphasize the possible link between hot spots and the mega-sized ULVZ. Thus, we added the mega-sized ULVZ beneath Samoa from Thorne et al. (2020; 2021) in the Introduction in Line 63 and also included other mega-sized ULVZs in later discussion.

Q8: Line 88: What are you calling δt_{ScS} and δt_S ? I assume you mean the difference of the ScS and direct S-wave arrival respectively with respect to some reference model such as PREM. But you don't actually tell us? I see that actually you make a statement in one of the figure captions that this may be referenced against IASP91. Please clarify in the text.

A8: We have clarified this part as " δt_{ScS} and δt_S are relative to IASP91 model" in Line 72.

Q9: Line 97: "For the northernmost event D, the change of δt_S is only up to 5 s, suggesting a contribution from the shallow structure beneath Alaska." I have no idea what this means. There is a 5 s change from event A also, and a 6 s change from the next event. Why does this particular event suggest shallow structure and the others do not?

Q10: Line 102: Here the authors state that they use the travel-time anomaly of SKS for event E to correct for upper mantle structure. This is completely opaque to as to why. The SKS path has almost no overlap with any of the modeled events, except perhaps for a small region immediately beneath Alaska. What thickness range beneath Alaska does this correct for? What do you do about the anomalous source-side of structure that SKS for this event encounters on its way down? I feel that this needs to be further justified, as to why this SKS path is representative of the upper mantle beneath Alaska and not just contaminated by other structure along the way.

A9-A10: Both comments are related to travel time correction for possible effects from shallow structures beneath Alaska. Considering the slab structure beneath Alaska, a proper correction of the shallow structure is needed. Here, we use SKS (sampling the receiver-side structure in a deeper angle) for event E and also include SS for the individual events (sampling the receiver-side structure in a shallower angle and shallower part of the lower mantle) to make the travel time corrections. Corrections with different reference phases (Fig. S3 and "S-wave travel time correction of events A-D" section in Methods) give similar trends, which rule out strong effects from the receiver-side structure and support our conclusion that observed the change of δt_S is driven by the lower mantle structure. The good consistency between corrections using different reference phases is true for all four events, which also indicates that source-

side structure has little effects on the corrected δt_s . Thus, we have re-organized and re-written this section as “Height of the Pacific LLVP”.

Q11: Line 108: “the onset distance of the δt_s jump”. It is not clear what this means. What δt_s jump are you referring to? Does this refer to the transition (such as shown in Fig. 2d) between positive and negative δt_s values?

A11: The “the onset distance of the δt_s jump” refers to “the distance of the initial increase of the δt_s along the N-S profile”, which is a good indication of S wave starting to encounter the top of the LLVP. Thus, such characteristic distance provides a robust constraint on the height of the LLVP. We have changed this sentence as in Line 97-Line 99.

In the original manuscript, Fig. 2a-b are for the N-S profile and Fig. 2c-d (plots for delayed and faster δt_s values) are for the W-E profile for Solomon Islands events recorded by North American stations. Thanks for pointing out the apparent confusion in this Figure. Thus, we have changed this old Fig. 2 into two figures. The new Fig. 2 focused on the N-S profile. And the new Fig. 3 focused on the W-E profile. We have also rewritten the text into two sections “Height of the Pacific LLVP” (N-S profile) and “Northern wall of the Pacific LLVP” (W-E profile) to describe in detail the edge structure of the Pacific LLVP.

Q12: Line 108: continuing the above point, the authors state this is where “...S is starting to sample the LLSVP which puts a good constraint on the height of the Pacific LLSVP.” It appears first of all that the largest anomalies are where the S-wave is faster (i.e., black open circles in Fig. 2d). Here S is advanced by as much as 2 s, whereas it appears to only be delayed by 1 s in the LLSVP region. In this case it seems that there is some fast anomaly that dominates the signal, and not necessarily the LLSVP. Also, I don’t see how much constraint is given on the height of the LLSVP through these data? The next sentence makes a statement trying to clarify this point, but perhaps drawing some ray paths would help to see how much height difference there is.

A12: The old Figs. 2c-d, now new Fig. 3, are for the W-E profile, which are used to determine the lateral extent of the LLVP. The height of the LLVP is mainly constrained from N-S profile as demonstrated in the new Fig. 2. We also include a new Supplementary Figure (Fig. S4) to demonstrate how the height of the LLVP affects the distance of the initial increase of the δt_s along the N-S profile. As displayed in Fig. S4, the distance of the initial increase of the δt_s gets smaller when the height of the LLVP is bigger.

Q13: Line 111: Now the authors immediately jump into their “preferred LLSVP model” but no information is given on how they arrive at this preferred model.

A13: We have rephrased this sentence in Line 101-Line 105. As discussed in the early section, the distance of the initial increase of the δt_s provides a robust constraint on the height of the LLVP as demonstrated in Fig. S4. Furthermore, by fitting δt_s , we can obtain the velocity structure of the LLVP. Here, we only assume a radial varied δV_s in the LLVP in this model with limited resolution. However, we include discussion in Line 121-Line 125 and Fig. S6 by having different LLVP geometry.

Q14: Starting at line 136: The authors jump in to stating a hybrid LLSVP model and a mega-sized ULVZ model are inadequate in explaining ScS data, but never do they really define what these models are. The hybrid LLSVP model is presumably the “preferred” model they constructed somewhere at some time, possibly in this paper. But what is the mega-sized ULVZ model they refer to?

A14: In the revised manuscript, we have specified our new 3D ULVZ model as NP ULVZ model in line 156. The mega-sized ULVZ model is for those identified ULVZ with large scales. Here we also renamed the mega-sized ULVZ model near Hawaii from Cottaar and Romanowicz’s (2012) as “HM ULVZ model” in Line 56 to make a better comparison.

Q15: Line 149: “A ULVZ model With 50 km height and -10% δV_s predicts S and ScS data for all events...”. OK, but is there not a fundamental tradeoff between and thickness for S and ScS such that a thinner ULVZ with strong δV_s might also work? The uncertainties and tradeoffs need to be discussed. It is impossible to tell from extended data figure 6 that another model doesn’t work.

A15: All three reviewers have pointed out the need for a quantitative assessment of the uniqueness of parameters for the NP ULVZ. Hence we have done extensive 2D, 3D trade-off tests on parameters of ULVZ models, i.e., δV_s , height, size, shape, density, and possible existence of heterogeneities. In particular, we use average cross-correlation coefficients as a criterion to grid-search δV_s and height for both 2D and 3D cases to determine the best combination of δV_s and height to best explain the waveforms of all events. We added a new section “Sensitivity tests on the model parameters” in the main text and describe more details in Supplementary Information S1 as a section of “Sensitivity tests for ULVZ models”. We also supply a new Table S3 to summarize how these parameters affect the waveforms of different events.

Q16: Fig 2. In panel b the y-axis goes up to 20 s. Perhaps this could be improved if it was cut down to 10 s. The observations aren’t larger than 10 s, so it might be easier to see the variation in the δt_s measurements if the y-axis scale is reduced.

A16: We have cut y-axis down from 20 s to 15 s for better visual inspection in new Fig. 2b, Fig. S3, Fig. S6.

Q17: Fig 3. In panel b, the synthetics actually look like they capture the nature of the data well. However, can something more quantitative also be shown, for example how much does the timing of the post-cursor differ for data and the (Tomo+LLSVP+megasized ULVZ) relative to the (Tomo+LLSVP+NP ULVZ) model? That is, I can't visually inspect these seismograms and say that oh yes, one model definitely performs better than another.

A17: We have supplied the average cross-correlation coefficients between data and synthetics in each waveform panel to show the goodness of fitting.

Review #2

Q1: In their explanations for the NP ULVZ, they suggest a less dense ULVZ because of a smaller δV_S of -10% compared with other mega-sized ULVZs. However, the authors did not take into account the density perturbations of ULVZ in their simulations. I understand the heavy computations required in a global 3D simulation, but it would be better if the authors could show how the density of ULVZ affects the synthetic waveforms in their sensitivity tests.

A1: We have calculated three 3D ULVZ models with +10%, -10% and +20% density perturbation with respect to PREM showed in Fig. S13K-M, respectively. All cases predict almost identical waveforms for both N-S and W-E profiles to those generating from the NP ULVZ model without perturbing the density except for ULVZ model with +20% predicts slight waveform amplitude variation, which indicates that the seismic phases at the period in this study (>10 s) have negligible sensitivity to the density.

Q2: The authors constructed the NP ULVZ model as a whole, is it possible that their ULVZ model consists of two individual patches of ULVZs, one is similar as the previously-inferred mega-sized ULVZ, the other one is a smaller-sized ULVZ. The author should exclude this possibility that the NP ULVZ is not composed of separated patches of ULVZs.

A2: Thanks for this interesting proposal. In Fig. S13N-S13Q, we test some ULVZ models with non-uniform δV_S or separated patches to examine how well they can predict the data. In one particular model (Fig. S13O), we used the mega-sized ULVZ from Cottaar and Romanowicz's (2012) plus a separated ULVZ model located at the west. Such model produces much stronger ScS delays at large azimuth even up to 27° for event B and much stronger Sdiff postcursors with long codas on profile WE. Our test results therefore conclude that NP ULVZ model represents the ULVZ structure better than separated patches.

Q3: I appreciate that the authors also analyzed the additional P-wave data to provide more constraints on the ULVZ properties. According to their PKP results, the $\delta V_P / \delta V_S$ of the NP ULVZ is about 1:2. However, the negative 5% of P wave velocity is not in agreement with results of Luo et al. (2001). If the result from Luo et al. (2001) is taken, the $\delta V_P / \delta V_S$ ratio of the ULVZ is 1:1, possibly suggesting a compositional origin. Thus I suggest the authors perform more detailed analysis of PKP data that approximately sample the same ULVZ region at the northern boundary of the Pacific LLSVP, because it is important in determining whether ULVZs represent patches of partial melt or iron-rich oxide. Based on this, the authors could shed more light on the origin of ULVZ in this area.

A3: We add more measurements of differential travel time residuals between PKPab and PKPdf ($\delta t_{PKP(ab-df)}$) for events along the Tonga-Fiji-Kermadec region recorded by

stations in Europe (Fig. S14), which sample similar region in Luo et al.'s (2001). However, our dataset shows that PKPab only displays ~1 s travel time variation and has no obvious waveform complexity. To match such PKP observations, a P-velocity drop less than 5% is sufficient for explaining δt_{PKPab} with ~1 s variation and generating no waveform distortion. The $\delta V_P/\delta V_S = 2$ for the NP ULVZ may suggest its compositional origin. We describe this point in detail in "P-velocity reduction of the NP ULVZ" section in the main text. We suspect that the observed waveform complexities and delayed PKPab in Luo et al.'s study may partly originate from sharp upper mantle structures, such as the slab, which requires more elaborate further work to confirm our point.

Q4: Could the authors use a quantitative measure of fit to determine the best-fitting models? Currently, the authors only compared the observations and synthetics visually. For example, in Fig. 3a, I cannot see many differences in the synthetics between LLSVP model plus mega-sized ULVZ and LLSVP model plus the preferred NP ULVZ. In contrast, in Fig. 3b, the synthetics from LLSVP plus mega-sized ULVZ model are visually better than those from LLSVP plus NP ULVZ model for the distance range of 50-65°.

A4: We have added cross-correlation coefficients between data and synthetics in each waveform panel to quantitative assessment of the uniqueness of parameters for the NP ULVZ. Thus, we have included a new section "Sensitivity tests on the model parameters" and series of tests in Supplementary Information to demonstrate how different model parameters affect the waveforms.

Q5: How did the authors choose the source time function in their simulation?

A5: We use a Gaussian function to approximate the source time function in both 2D and 3D simulations using the half duration given in GlobalCMT. Although such a simple source time function may not represent exactly the earthquake source, the generated synthetics predict the waveforms at all distances well at the period in our study.

Q6: It would be better to add the Pearson's r-value in the Extended Data Fig. 1, so that people could know how the results improve after the corrections.

A6: Thanks for this suggestion. We have added the Pearson's R-value in the Fig. S1.

Q7: Line 102-104: The authors used δt_{SKS} from event E to correct for upper mantle structure beneath Alaska. According to the raypath of SKS, how would the author rule out the effects from source-side heterogeneity within LLSVP on the δt_{SKS} when the raypaths of SKS pass through?

A7: We have replied these comments partially in review 1's Q9-Q10. In the revised manuscript, we also use SS, which has ray paths almost identical in the uppermost mantle beneath the source and station of S, to make the travel time correction of S. We describe this in detail as for the "S-wave travel time correction of events A–D" in the "Methods" section. The good consistency between corrections using different reference phases is true for all four events (Fig. S3), which indicates that the source-side structure has little effect on the corrected δt_s .

Q8: I am a little skeptical about the inferred heights of the authors' preferred LLSVP, if the authors cannot take into the upper mantle heterogeneities properly. The used five events are widely distributed, how did the authors exclude the effects on δt_s from upper mantle heterogeneities in the source regions when considering the new LLSVP model? How did the authors calculate the S wave travel times in 3D models, based on 1D or 3D ray tracing methods? In Fig. 2d and Extended Data Fig. 3, the upper height of the colored circles is less than the top height of the preferred LLSVP (purple line), then how would the authors infer that the LLSVP model has the height of 920 km?

A8: We excluded the upper mantle heterogeneities in the source regions as described above in A7. We calculate the S wave travel times in 3D models based on 3D ray tracing methods to correct travel time residuals. Sorry that we generated confusion in the old Fig. 2 by combining the results from N-S profile (old Fig. 2a-b) and W-E profile (old Fig. 2c-d). In the revised manuscript, we have re-organized them into two separated figures (Fig. 2 and Fig. 3). In the old Fig. 2d (new Fig. 3), we measure the δt_s on a W-E profile with events in the Solomon Islands and records in North America and project δt_s at their turning depth of S rays on N-S profile. Thus, the "upper height of the colored circles" does not provide information about the height of the LLVP. However, their sharp lateral changes provide good constraints on the north edge of the LLVP. In fact, the height of the LLVP is mainly derived from the distance of the initial increase of the δt_s along the N-S profile (new Fig. 2).

Q9: The authors use the PKPab and PKPdf data to further constrain the P wave velocity of NP ULVZ. Which frequency band did the authors use to filter the PKP data? The author should show an example of observed PKP seismograms as well as comparison between the observations and synthetic waveforms based on the NP ULVZ model.

A9: The PKP data are filtered to 1–50 s. We have included a description in the caption of Fig. S14. We also added synthetics and example waveforms of PKPdf and PKPab for event 20180930 with PKPab ray paths sampling the NP ULVZ region (Fig. S14c-d).

Q10: How would the authors evaluate the effects from the receiver-side structures beneath the Europe when they calculate the PKP differential travel time residuals?

A10: This is a great point. Our argument for the strong PKPab delays and waveform distortions observed in Luo et al. (2001) may well be affected by some very local structures beneath Europe, in particular Spain. As displayed in Fig. S14a, the ray paths of PKPab and PKPdf are in fact quite different at the shallow mantle. Unfortunately, due to computation limit, we can now only generated 2D synthetics for PKP assuming the GyPSuM model, which might have a low-resolution upper mantle beneath Europe. Hence this will be a future research avenue to fully understand how the upper mantle affects the PKP behaviors.

Q11: The authors measure the differential time residuals relative to PREM for PKP, but for ScS-S measurement, they use iasp91 model as reference. I am wondering why the authors did not use the same model for consistency.

A11: We have recalculated the differential time residuals relative IASP91 for PKP data as displayed in Fig. S14.

Q12: The filter bands for S waves are sometimes confusing. In the main text, the filter band is 10-50s, but in the Supplementary Information, the filter band is 5–50s (Line 38), please make them consistent.

Q13: In Fig. 3, why did the author choose a different filter band (10-20 s) for the event F?

A12-A13: We generate all synthetics at a broad frequency band. In 2D simulations, we select the filter band to 5–50 s for events A–D, which allows a better resolution of the structure. For 3D simulations, due to the heavy computation, the shortest period for event B is at ~8 s, so we filter data to 10–50 s. For event F, we also filter data to 10–20 s, which is used in Cottaar and Romanowicz (2012), for a better comparison. We describe this in Supplementary Information S1.

Q14: Figure 4 b-d look just fine for explaining the accumulation and fragmentation of the NP ULVZ. The authors could have plotted a more beautiful figure.

A14: We have improved Fig. 4. Hope it now provides a better illustration.

Q15: Line 87-88: Please refer to the model through which the authors compute the travel time residuals.

A15: We have clarified this part as “ δt_{ScS} and δt_S are relative to IASP91 model” in Line 72.

Q16: In Line 86, the authors state that, the epicentral distances are smaller than 85° for ScS time residuals, however, in Line 90, the distance for δt_{ScS-S} is from 75° to 90° , which is confusing. Please clarify this.

A16: Here, we measure the δt_S and δt_{ScS} at distance smaller than 85° , at which S and ScS are well separated, to generate a composited travel time plots in Fig. S1 for all events with same selection criterion. For event B with high-quality data, we are able to extend the δt_{ScS-S} measurements to 90° , which are mainly modelled in this study. We have added a description in “Data processing procedures and travel time measurements” section in “Methods”.

Review #3

Q1: This paper presents a very nice data set, and a fair amount of modeling. The topic is important. In my opinion, for the paper to be suitable for a high profile journal, something new and novel of impact needs to be presented. This paper attempts making the case that a very large (mega) ulvz is broken into pieces. If that were indeed demonstrated, and a robust demonstration as to what it means, then I agree it is impact science. However, I do not agree that that conclusion is constrained at all. Even if the modeling were better explained (I think I figured out what was being done), the trade-offs are too large. The assumed constant property ulvz is unrealistic, especially to hold that fixed and then model shapes, when the shear velocity drop is relatively weak. If the CMB started with ulvz typical shear drops (e.g., 30%) and then transitioned to smaller amounts, what happens? For something that large, one does not expect constant property. I also struggled with visual side by side comparisons between data and synthetic seismograms. That is quite qualitative. I see many features suggesting fits are not that perfect, which signals trade-offs are possible. Only map view structural differences are shown, but that trades off, obviously, with vertical extent and properties, which are discussed faintly. In summary regarding the seismology, I think the data look nice, the modeling looks nice, but I do not see demonstration that the preferred structures are so unique. There is no compelling demonstration that the data are not affected by the source region (since waves leave similar to trench strike, which often results in slab multipathing). These are my first and biggest issues.

A1: Thanks for the encouraging comments on the data and modelling, which stimulated our large effort in the revised manuscript to address the issues of uniqueness and trade-offs (also see our responses to Reviewers #1 and #2). Adopting the principle of Occam's razor, it is common practice in seismic waveform modelling to assume relatively simple geometries with uniform physical properties. Ultimately, this choice is guided by the data available which prevents characterization of deep Earth structures at the same resolution as the crust or upper mantle. To reiterate, a main contribution of our study is including two orthogonal profiles (N-S and W-E profiles) to break the degeneracy between the size and velocity perturbation of a ULVZ. Following the suggestions of all reviewers, we have performed extensive trade-off tests (Figs. S8-S9, S12) as well as sensitivity tests on various parameters of the ULVZ, i.e., δV_s , height, size, shape (Fig. S13). In these tests, we use cross-correlation coefficients (CC's) between data and synthetics as a criterion to judge the goodness of waveform fit quantitatively, rather than only visual inspection. We appreciate all reviewer's comments on the quantitative assessment to better demonstrate how these different models impact 3D wave propagation. We also examine models including heterogeneities, separated ULVZ patches, and a gradual radial velocity perturbation as suggested. The synthetics tests suggest that models with separated ULVZ patches cannot predict the data well (Fig. S13). Our extensive series of new tests provide yet more evidence that the NP ULVZ model as a whole describes well the ULVZ structure at the northern edge of the Pacific LLVP.

Q2: My second issue concerns interpretation. Flow model implications are important, but I struggle to believe any particular flow model because all of the assumptions involved. I think that citing dynamics flow behavior from a convection model without connection to any particular geography seems okay for wisdom on general flow behavior (and seems to be done in papers), but tying a particular flow to some specific spot requires a lot of faith. In fact, the same goes with tomography – discussing subvertical walls... those types of features may be very model dependent, depending upon the location. It also depends upon path coverage (both the African and Pacific LLSVPs have tilts along raypaths). One model was chosen, which is referenced for this kind of interpretation work a lot less than Ritsema or French & Romanowicz. There are many papers discussing the variability of models. So I feel the dynamics and reference to tomography details too speculative. A lot of adjectives in description of dynamics were excessive and unfounded/undemonstrated in imaging studies. E.g., line 226 “vigorous” flow, “rapid” southward motion. Later, ‘slab debris deforming the LLSVP wall’ is also pure speculation. There is a lot of that in this paper, and it is pure speculation. The argument for two different territories (line 234) is also unfounded, since the Iceland structure was modeled solely with diffracted waves (thus low resolution). Thus, while I do appreciate the nice data and hard work of the modeling, I do not think the conclusions and preferred models are constrained.

A2: An objective is to explain the observed different ULVZ structures at the northern edge (enormous NP ULVZ) and the northeastern edge (fragmented small-size ULVZs) of the Pacific LLVP, which are affected by different types of flow introduced by the slab. Firstly, in the tomographic model, distinct high-velocity regions at the CMB are situated at the northern and northeastern edge of the Pacific LLVP (Fig. 1b). He and Wen (2012) also confirmed that D'' discontinuity locate just north of the inferred edge of the north Pacific LLVP in our study. We added this description in the “Discussion” section in the main text. Both observations indicate the possible existence of slab debris. Then, we associate the observed slab debris with results from geodynamic models that utilize plate reconstructions (e.g., Bower et al., 2013; Hassan et al., 2016) and show a stable convergent flow from the north. Therefore, the convection models we appeal to are directly connected to the geography of the region that we study.

Secondly, the subvertical northern wall may be also a good indication of a “slab push” at the northern edge of the Pacific LLVP. Here, we draw the conclusion of the subvertical northern wall purely by fitting the ScS-S differential travel time along a N-S profile and the azimuthal jump of S travel times along the W-E profile (new Fig. 2-3), which is independent of tomographic models. As the main text describes, a subvertical wall constrains the nature of geodynamic flow in this region since a dense LLVP would form inward sloping boundaries (McNamara and Zhong 2005, Tan and Gurnis 2007, Ming Ming Li et al. and others). Hence the LLVP is either less dense/neutrally buoyant and/or a slab must be impinging to steepen the edge. Crucially, we appeal to results from geodynamic modelling of several groups over several years to inform our interpretation.

Therefore, we have revised the discussion of the connection between the seismology and geodynamic interpretation to focus on these primary results. We have removed adjectives that imply speculation when in fact our interpretation is guided by previous modelling efforts. Ultimately, we are proposing a reasonable geodynamic scenario based on new seismic data, and this testable hypothesis can be interrogated with the next generation of geodynamic models.

Upon reassessment, we agree that the argument for two different territories is at present too ambitious and will require more work to confirm or refute. Thus, we have removed such comparison between Hawaii and Iceland, considering the resolution with diffracted waves only. In the revised manuscript, we have reorganized and rewritten the discussion by focusing on explaining the large scale of the NP ULVZ and the difference between ULVZs at the northern and northeastern edge of the Pacific LLVP.

REVIEWER COMMENTS

Reviewer #2 (Remarks to the Author):

The authors have addressed major points that I raised during the first round of review. In particular, they have computed cross-correlation coefficients between data and synthetics to quantitatively assess the uniqueness of parameters for the NP ULVZ, such as density, shape, and trade-offs between velocity and height. I appreciated their extra work for the 3D simulations. Now the main concern for me is, since the authors claim the NP ULVZ is enormously large even compared to several previously detected mega-sized ULVZs, I think the authors need to explain more specifically to the readers about the special reasons (maybe geologically or dynamically or mineralogically) for such a unusually large ULVZ in North Pacific.

Some other comments/suggestions:

1. In the discussion section, the authors mentioned the stable convergent mantle convection. I think it would be better if the authors could explain more on what is the stable convergent mantle convection? Why is it stable?
2. In the section "the height of the Pacific LLVP", the authors need clarify the approach in obtaining the velocity profile mentioned in Line 103-104. Is it based on an inversion method? The authors only say "match the corrected δt S well", it is not very clear how the authors calculate the velocity profile.
3. The authors just mentioned the chemical origin for the ULVZ based on the ratio of $\delta V_s/\delta V_p$ being ~ 2 . More discussions on the possible compositional origin of ULVZ could be added.
4. Some extra synthetic tests for the effect of the CMB topography on the waveforms would strengthen the authors' conclusion, since such a large denser ULVZ would probably result in some variations of the CMB topography.
5. Fig. 5 b-d could have been plotted nicer. The current Fig b-d look just ok, but are not of high quality to demonstrate the possible dynamic processes.
6. Fig. S6: It looks like the LLVP has a south tilting edge which is not consistent with that in Fig. S3.

Line 36: at large scale => at large scales

Line 80,95: I suggest the authors using S waves instead of S.

Line 189: in CMB size => at the CMB

Line 254: likely be => likely to be

Reviewer #3 (Remarks to the Author):

The authors have revised their manuscript and made changes in attempts to address concerns of all three reviewers. My positive comments still hold from my last review. My concerns about interpretation still hold, but the authors attempted to justify their position. However, the reality is that it remains unknown for some regions surrounding LLVPs if the flow is towards or away from the LLVP because we only have today's snapshot. Just ask a global 3D geodynamicist that works on thermochemical piles. Nonetheless, seismology based papers routinely make such claims whether or not they are well constrained. So, while my preference is to temper the interpretation more as hypothesis, I can live with it.

I was less compelled by the arguments about the travel times and their trade-offs raised by reviewers and the responses. I will leave that up to the other reviewers to respond to the changes made. For sure there are large uncertainties due to heterogeneity away from the region of interest. But the authors down play those with confidence I don't totally share. Maybe the other reviewers are happy with the changes made.

I do not really agree with the authors' response arguing an "Occam's razor" philosophy for fixing the ULVZ velocity. One could make the same argument for LLVPs: maybe tomographers should fix the LLVP velocity then model the shape. But dV varies significantly within LLVPs. The same will be true for ULVZ. On the other hand, if the wave path coverage or modeling methodology does not have constraints on such variability, then say it, acknowledge the trade off. A test or two can be done to model variable dV within ULVZs, and see: how do predictions vary?

Cross-correlation is a step in the right direction for comparing data and synthetics. However, there is no detail here. CC is a tool that cares most about the largest amplitudes. It does not distinguish between waveform subtleties. Are the entire waveforms correlated? If yes, then it is really unclear what the CC coefficient provides. Are the phases of interest windowed? Or just ScS? If the latter, is the time shift used in assessing best model? Some record sections show interference with the depth phase (I assume, sS), and should not be included in the correlation. If you are correlating the whole trace then travel times are dominating the CC's. If you are correlating just ScS, then wave shapes are dominating the CC's. To me, the authors have not demonstrated how they are using CC, nor how their answers depend on it. Another way to do this is to use some kind of measure of misfit of the data and synthetics (but only do it for the parts of the waveform that matter, e.g., ScS, where it's time is set from aligning the S waves). A common way is to subtract one record from the other, then integrate the remainder trace for a proxy estimate of energy of misfit between the records. I feel that is better than CC. But anyway, the tool needs to be defined, and shown to matter. I do not feel this is a deal breaker, because the authors can do it.

I struggled with the new language of "enormous". First, it is just another very large ULVZ, so they can use the "mega" terminology set up in previous work. (however, I still do not think they can rule out closely located smaller things). That might set up a challenge because it, in principle, has been put forth in a small handful of other papers (i.e., not conceptually a discovery). However, they argue it is only mild in velocity drop. That is new. But if small ULVZs are twice the shear reduction, and convection sweeps them to a pile margin (where it is hotter), how does the shear drop become LESS? Second, using the article "the" before "enormous" implies definitive, as if it is something that is unique on the planet, which of course the study cannot claim. Perhaps instead the authors can take the perspective that another mega ULVZ is found, but a new kind (bigger, less anomalous). and thus mega ULVZs may not be rare. But why is it less anomalous? They can then formulate some hypothesis about so much ULVZ volume (with property diversity) that is novel. Why are there huge ones and small ones? Why weak versus strong? I would go to geodynamics for information on size (Mingming Li's papers are probably a good start - he's done a lot on the morphology). But shearing a piece of enormous ulvz to make little ones, and have them be twice as anomalous? I just don't see the logic.

So, as it stands, the paper improved. But I still struggle with the trade offs which the authors didn't fully address in my opinion, the synthetic/data comparison measurements (how is CC done? How good is it as a measure of data/synthetic fit), and the logic around the hypotheses that are presented. But I would like to close by encouraging the authors to rework these issues because it has ingredients of being important work that will impact a broad and diverse readership. I appreciate the hard work they are putting into this study.

Reviewer #4 (Remarks to the Author):

Summary

This study uses differential travel times between S and ScS waves to investigate the properties of an Ultra-Low Velocity Zone (ULVZs) above the core-mantle boundary in the north Pacific. Specifically, they use signals from several earthquakes located in the southwest Pacific recorded on the USArray to sample the core-mantle boundary near the northern edge of the Pacific LLVP from multiple azimuths. They confirm the presence of a ULVZ at the core-mantle boundary, which they call the Northern Pacific ULVZ, and find that, compared to previous studies (e.g., Cottaar and Romanowicz, 2012), the ULVZ is elongated in the E-W direction, and is

characterized by a weaker shear velocity anomaly ($dVs = -10\%$). In general, the paper is well written and will be of interest to researchers across multiple disciplines. The seismic observation and modeling appears robust, however, I find discussion section of the paper to be highly speculative and unsupported by the results. Specifically, I do not see how the proposed model of ULVZ fragmentation, for which the paper is named, is confirmed by the observations. From what I can tell, the results improve the constraints on the properties of the northern Pacific ULVZ, but the proposed model requires knowing the mantle flow history at the core-mantle boundary, which cannot be gleaned from the data presented in this study. I would recommend that the authors word their interpretation of their results a little less strongly, and perhaps provide some possibilities for how future studies may support their proposed model. Additionally, I have several specific comments that I would like the authors to address prior to publication.

Specific Comments

- 1) Figure 1. I would suggest adding a scale bar in km to panel B so that people have a better sense of the sizes of the ULVZs at the CMB.
- 2) Figure 2a. While the text explains that the cross section is taken between Tonga and Alaska, the exact path is not shown. Perhaps you could draw the cross section track in panel a of Figure 1.
- 3) Line 55: "Cottaar and Romanowicz propose a strikingly large-sized ULVZ near Hawaii (referred to as the Hawaiian mega-sized (HM) ULVZ hereafter in this paper)" Later in the paper, this ULVZ is not consistently referred to as the HM ULVZ. It is referred to either as the "mega-sized ULVZ" or "HM ULVZ". It would be easier for readers if the authors kept things consistent.
- 4) Line 125: "In contrast, an LLVP with a south tilting edge and a wider base produces stronger delays at larger distance than data..." Should this read "produces stronger delays at a larger distance than observed in the data"?
- 5) The authors performed a detailed sensitivity analysis to explore the tradeoff between the height and shear velocity anomaly of the ULVZ, which I appreciated. However, I wish that the results were used to discuss the uncertainty on their preferred model. The authors state that a model with $dVs = -10\%$ fits the data the best based on correlation coefficient, but as shown in the Supplemental Material (e.g. Fig. S9) there is a strong tradeoff between parameters, and there exists a region of plausibly acceptable models. I would suggest using the misfit surface to assess the model uncertainty.
- 6) Line 133: "A two-dimensional (2D) ULVZ model spanning 12° in the N-S direction with 50 km height and $-10\% \delta VS$ predicts S and ScS data for all events A-D along this particular corridor (Figs. S8-S9)." I think many readers will find a "2D" model confusing. I understand that the authors performed both 2D and 3D numerical simulations, referring to the ULVZ as a 2D structure doesn't make sense to me. The 2D simulations are simply taking a slice through the 3D ULVZ.
- 7) It seems like some important results are being relegated to the Supplemental material. For example, the results shown in Figs S1 and S2 are the first results discussed in the paper and may be useful for the readers to understand the context of the study.

Dear Reviewers,

We thank you for your great and constructive comments.

The quotes from your comments are followed by our responses in *blue italics*.

Reviewer #2

Q1: The authors have addressed major points that I raised during the first round of review. In particular, they have computed cross-correlation coefficients between data and synthetics to quantitatively assess the uniqueness of parameters for the NP ULVZ, such as density, shape, and trade-offs between velocity and height. I appreciated their extra work for the 3D simulations. Now the main concern for me is, since the authors claim the NP ULVZ is enormously large even compared to several previously detected mega-sized ULVZs, I think the authors need to explain more specifically to the readers about the special reasons (maybe geologically or dynamically or mineralogically) for such an unusually large ULVZ in North Pacific.

Q2: In the discussion section, the authors mentioned the stable convergent mantle convection. I think it would be better if the authors could explain more on what is the stable convergent mantle convection? Why is it stable?

A1-A2: We have included more discussion about why the NP ULVZ has a large size. In terms of geodynamics, a long-term and stable convergent mantle flow in this region is derived from slab subduction in the northern Pacific. The slabs establish a regional mantle wind that causes the accumulation of the ULVZ materials at the northern edge of the LLVP. The establishment of long-lived and rounded ULVZs from “stable convergent mantle convection” is inspired by the geodynamic simulations in Li et al., 2017 (Ref. 39). Although there are uncertainties in both geodynamical and seismic modelling, the NP ULVZ model together with the patchy ULVZs at the northeastern edge of the Pacific LLVP replicate features in Li et al’s geodynamic models. Hence we appeal to these geodynamic simulations to connect a “stable convergent mantle convection” to the development of a mega-sized ULVZ. In the discussion, we also discuss the chemical properties of the NP ULVZ since $\delta V_S/\delta V_P$ is approximately 2, as well as the effects of the possible existence of a “broad plume”.

Q3: In the section "the height of the Pacific LLVP", the authors need clarify the approach in obtaining the velocity profile mentioned in Line 103-104. Is it based on an inversion method? The authors only say “match the corrected δt_s well”, it is not very clear how the authors calculate the velocity profile.

A3: Because the height of LLVP is robustly determined by the distance of the initial increase of δt_s (new Fig. S3), we can fix the height of the LLVP. Then, we use trail-and-error approach to find the δV_s , which can best describe the δt_s of all four events. We have included more description at Lines 99-103. In contrast, we also show an LLVP with a south tilting edge and a wider base, which produces stronger delays at large distance than observed in the data, in particular for event A at distance larger than 90° (new Fig. S5).

Q4: The authors just mentioned the chemical origin for the ULVZ based on the ratio of $\delta V_s/\delta V_p$ being ~ 2 . More discussions on the possible compositional origin of ULVZ could be added.

A4: We have added more discussions on the possible compositional origin of ULVZ at Lines 239-245.

Q5: Some extra synthetic tests for the effect of the CMB topography on the waveforms would strengthen the authors' conclusion, since such a large denser ULVZ would probably result in some variations of the CMB topography.

A5: Thank you for this suggestion. Due to the difficulties of implementing CMB topography in SPECFEM 3D simulations, we limit our discussion on the 2D CMB topography. In Fig. S16, models with CMB topography up to 20 km are examined. Although the modelled CMB topography is quite large, our synthetics show that such models do not have significant effects on both amplitude and travel time of the N-S profile as the NP ULVZ model. We have added discussion on the CMB topography at Lines 204-207.

Q6: Fig. 5 b-d could have been plotted nicer. The current Fig b-d look just ok, but are not of high quality to demonstrate the possible dynamic processes.

A6. Following suggestions from reviews #3 and #4, we have deleted the original Fig. 5d to emphasize more the effect of stable convergent flow on forming the mega-sized ULVZ with new Fig. 6b-c. We explain the reason for mega-sized and small patchy ULVZs in the text. We have also synchronized the colors in new Fig. 6b-c with Fig. 6a.

Q7: Fig. S6: It looks like the LLVP has a south tilting edge which is not consistent with that in Fig. S3.

A7: Yes, you are right. The model in old Fig. S3 (new Fig. S2) is our preferred LLVP model with a north tilting edge. In old Fig. S6 (new Fig. S5), we set up a model with a south tilting edge and examine how the different wall will affect the δt_s . As shown in

new Fig. S5, an LLVP with a south tilting edge and a wider base produces stronger delays at larger distance than data, in particular for event A at distance larger than 90° . Hence a north tilting edge fits data better than a south tilting edge.

Reviewer #3

Q1: The authors have revised their manuscript and made changes in attempts to address concerns of all three reviewers. My positive comments still hold from my last review. My concerns about interpretation still hold, but the authors attempted to justify their position. However, the reality is that it remains unknown for some regions surrounding LLVPs if the flow is towards or away from the LLVP because we only have today's snapshot. Just ask a global 3D geodynamicist that works on thermochemical piles. Nonetheless, seismology based papers routinely make such claims whether or not they are well constrained. So, while my preference is to temper the interpretation more as hypothesis, I can live with it.

A1: We agree that it is difficult to fully understand the lower mantle flow patterns, which are often based on geodynamic simulations under certain assumptions at present day. In the revised manuscript, we have reorganized the discussion and focused more on seismological observations. Then we discuss the similarity between our observation and the advanced dynamic model from Li et al (2017) (Ref. 39), which supports the hypothesis that the mantle flow has strong effects on developing different geometries of the ULVZ.

Q2: I was less compelled by the arguments about the travel times and their trade-offs raised by reviewers and the responses. I will leave that up to the other reviewers to respond to the changes made. For sure there are large uncertainties due to heterogeneity away from the region of interest. But the authors down play those with confidence I don't totally share. Maybe the other reviewers are happy with the changes made.

A2: With limited ray coverage, it is difficult to completely resolve this issue. Here, we argue that the height of the LLVP is a robust feature. The δV_S in the LLVP has a trade-off with the lateral extension of the LLVP if only fitting the travel time along the N-S profile. However, with the extra constraint from the W-E profile, we can improve the definition of the northern edge of the LLVP, in particular its slope.

Q3: I do not really agree with the authors' response arguing an "Occam's razor" philosophy for fixing the ULVZ velocity. One could make the same argument for LLVPs: maybe tomographers should fix the LLVP velocity then model the shape. But dV varies significantly within LLVPs. The same will be true for ULVZ. On the other hand, if the wave path coverage or modeling methodology does not have constraints on such variability, then say it, acknowledge the trade off. A test or two can be done to model variable dV within ULVZs, and see: how do predictions vary?

A3: To mimic variable δV_S within the NP ULVZ, we develop some tomography-like

models by enhancing original global tomographic models (new Fig. S15U-W) as well as models with a gradient of δV_S in depth (new Fig. S16R) or small-scale lateral heterogeneities (with correlation length of ~ 60 km) (new Fig. S15T). We also include two heterogeneous ULVZ models by varying either δV_S or height to interpret the observed $\delta t_{\text{ScS-S}}$ along the the N-S profile (new Fig. S15X-Y). Based on the 3D synthetics for these models, (1) Fig. S15R-S implies that the radial variation of δV_S in NP ULVZ should not exceed 20%, (2) Fig. S15T-Y implies that heterogeneous ULVZ models with a mean δV_S of -10% predict the data well. However, the lateral δV_S variation should not exceed 4% at the wavelength of ~ 200 km, which is compatible to the Fresnel Zone. In Fig. S15Z, we also add a ULVZ model composed of small patches with different sizes and stronger velocity perturbation, which fails to predict the data. These tests suggest that the NP ULVZ is a continuous block at long wavelength rather than many small, disconnected patches. We acknowledge smaller and stronger variations might exist at a level below the seismic detection threshold, but our uniform NP ULVZ model represents the bulk properties of the ULVZ, which can best fit currently available data from the N-S and W-E profiles. We have added discussion in the section of “Sensitivity tests on the ULVZ model parameters”.

Q4: Cross-correlation is a step in the right direction for comparing data and synthetics. However, there is no detail here. CC is a tool that cares most about the largest amplitudes. It does not distinguish between waveform subtleties. Are the entire waveforms correlated? If yes, then it is really unclear what the CC coefficient provides. Are the phases of interest windowed? Or just ScS? If the latter, is the time shift used in assessing best model? Some record sections show interference with the depth phase (I assume, sS), and should not be included in the correlation. If you are correlating the whole trace then travel times are dominating the CC's. If you are correlating just ScS, then wave shapes are dominating the CC's. To me, the authors have not demonstrated how they are using CC, nor how their answers depend on it. Another way to do this is to use some kind of measure of misfit of the data and synthetics (but only do it for the parts of the waveform that matter, e.g., ScS, where it's time is set from aligning the S waves). A common way is to subtract one record from the other, then integrate the remainder trace for a proxy estimate of energy of misfit between the records. I feel that is better than CC. But anyway, the tool needs to be defined, and shown to matter. I do not feel this is a deal breaker, because the authors can do it.

A4: Now, in the “Methods”, we have included a section of “Misfit between data and synthetics” with a detailed description of the CC. For the N-S profile, we align waveforms on S peaks and select a time window of 20 s before S and 20 s after ScS predicted by IASP91 to perform CC. For the W-E profile, the time window is fixed as 20 s before and 80 s after Sdiff to include the postcursors. Gray shaded areas are also added in Fig. 5 to display the time windows used to calculate CC. Note that both ScS and Sdiff postcursors are quite strong. Thus, the CC are not dominated by S or Sdiff. In addition, as suggested, we also calculate L1-norm and L2-norm (Methods) within the

same time window of calculating CC as the misfit function. We add a new Fig. S7 to show a comparison among three different misfit functions. In new Figs. S12-S14, we check all three misfit functions to examine the trade-offs between δV_s and H of the ULVZ and find the L1/L2-norm misfit functions give similar results as CC, implying the CC values used in this study are robust.

Q5: I struggled with the new language of “enormous”. First, it is just another very large ULVZ, so they can use the “mega” terminology set up in previous work. (however, I still do not think they can rule out closely located smaller things). That might set up a challenge because it, in principle, has been put forth in a small handful of other papers (i.e., not conceptually a discovery). However, they argue it is only mild in velocity drop. That is new. But if small ULVZs are twice the shear reduction, and convection sweeps them to a pile margin (where it is hotter), how does the shear drop become LESS? Second, using the article “the” before “enormous” implies definitive, as if it is something that is unique on the planet, which of course the study cannot claim. Perhaps instead the authors can take the perspective that another mega ULVZ is found, but a new kind (bigger, less anomalous). and thus mega ULVZs may not be rare. But why is it less anomalous? They can then formulate some hypothesis about so much ULVZ volume (with property diversity) that is novel. Why are there huge ones and small ones? Why weak versus strong? I would go to geodynamics for information on size (Mingming Li’s papers are probably a good start – he’s done a lot on the morphology). But shearing a piece of enormous ulvz to make little ones, and have them be twice as anomalous? I just don’t see the logic.

A5: This feedback has helped us to further frame our study. Indeed, several previous imaged mega-sized ULVZs are about the same size as the NP ULVZ. We have changed “enormous” to “mega-sized” throughout the manuscript to be consistent with previously introduced terminology. Regarding the difference between the NP ULVZ and other mega-sized ULVZ, we have reorganized and rewrote the “Discussion”. We hypothesize that a long-term, stable, horizontally convergent mantle might be the key to generating large sized ULVZ by connecting our work to Mingming Li’s (Ref. 39) paper. We have added discussion about possible different chemistries among different ULVZs. With different chemistry, density, and viscosity, ULVZs should have great diversity in geometry considering the effects from horizontal mantle flow as well as a “broad plume” suggested by (French and Romanowicz, 2015). We have included discussions regarding all these points.

Q6: So, as it stands, the paper improved. But I still struggle with the trade offs which the authors didn’t fully address in my opinion, the synthetic/data comparison measurements (how is CC done? How good is it as a measure of data/synthetic fit), and the logic around the hypotheses that are presented. But I would like to close by

encouraging the authors to rework these issues because it has ingredients of being important work that will impact a broad and diverse readership. I appreciate the hard work they are putting into this study.

A6: We appreciate these constructive comments and have further improved the manuscript accordingly.

Reviewer #4

This study uses differential travel times between S and ScS waves to investigate the properties of an Ultra-Low Velocity Zone (ULVZs) above the core-mantle boundary in the north Pacific. Specifically, they use signals from several earthquakes located in the southwest Pacific recorded on the USArray to sample the core-mantle boundary near the northern edge of the Pacific LLVP from multiple azimuths. They confirm the presence of a ULVZ at the core-mantle boundary, which they call the Northern Pacific ULVZ, and find that, compared to previous studies (e.g., Cottaar and Romanowicz, 2012), the ULVZ is elongated in the E-W direction, and is characterized by a weaker shear velocity anomaly ($dVs = -10\%$). In general, the paper is well written and will be of interest to researchers across multiple disciplines. The seismic observation and modeling appears robust, however, I find discussion section of the paper to be highly speculative and unsupported by the results. Specifically, I do not see how the proposed model of ULVZ fragmentation, for which the paper is named, is confirmed by the observations. From what I can tell, the results improve the constraints on the properties of the northern Pacific ULVZ, but the proposed model requires knowing the mantle flow history at the core-mantle boundary, which cannot be gleaned from the data presented in this study. I would recommend that the authors word their interpretation of their results a little less strongly, and perhaps provide some possibilities for how future studies may support their proposed model. Additionally, I have several specific comments that I would like the authors to address prior to publication.

A: In the revised manuscript, we have reorganized the “Discussion” and put less emphasis on ULVZ fragmentation. Instead, we focus on the NP ULVZ itself and add more discussion on mechanisms of forming such a large-size ULVZ, as suggested by all three reviews. Thus, we also change the title to “Slab Control on the Mega-sized North Pacific Ultra-low Velocity Zone”. In particular, following the geodynamic modelling work in Li et al. (2017) (Ref. 39), we hypothesize that stable convergent flow in the lower mantle beneath the north Pacific can drive the growth of large ULVZs. However, as pointed out by reviewers, accurate mantle flow patterns have not been resolved for this specific region. Therefore, obtaining such flow patterns is important for understanding the evolution of the ULVZ and its interaction with the LLVP in future studies. We have added discussion about this in the main text.

Specific Comments

Q1: Figure 1. I would suggest adding a scale bar in km to panel B so that people have a better sense of the sizes of the ULVZs at the CMB.

A1: We have added a scale bar in Fig. 1b.

Q2: Figure 2a. While the text explains that the cross section is taken between Tonga and Alaska, the exact path is not shown. Perhaps you could draw the cross section track in panel a of Figure 1.

A2: We have added a cross section track in Fig. 1a.

Q3: Line 55: “Cottaar and Romanowicz propose a strikingly large-sized ULVZ near Hawaii (referred to as the Hawaiian mega-sized (HM) ULVZ hereafter in this paper)” Later in the paper, this ULVZ is not consistently referred to as the HM ULVZ. It is referred to either as the “mega-sized ULVZ” or “HM ULVZ”. It would be easier for readers if the authors kept things consistent.

A3: We have used “HM ULVZ” throughout the new manuscript.

Q4: Line 125: “In contrast, an LLVP with a south tilting edge and a wider base produces stronger delays at larger distance than data...” Should this read “produces stronger delays at a larger distance than observed in the data”?

A4: We have changed this sentence.

Q5: The authors performed a detailed sensitivity analysis to explore the tradeoff between the height and shear velocity anomaly of the ULVZ, which I appreciated. However, I wish that the results were used to discuss the uncertainty on their preferred model. The authors state that a model with $dV_s = -10\%$ fits the data the best based on correlation coefficient, but as shown in the Supplemental Material (e.g. Fig. S9) there is a strong tradeoff between parameters, and there exists a region of plausibly acceptable models. I would suggest using the misfit surface to assess the model uncertainty.

A5: Following comment from review #2, in addition to calculating cross-correlation coefficient (CC), we also calculate L1-norm and L2-norm misfits between data and 3D synthetics to study the trade-offs between δV_s and H of the NP-ULVZ (new Figs. S12-S14). Simulations suggest that a trade-off exists between δV_s and height of the NP ULVZ (new Fig. S9) and we clarify the δV_s uncertainties in the ULVZ. From new Fig. S9, we conclude that a model with δV_s of $-10\% \pm 2\%$ and H of 50 ± 20 km can provide an overall good fit to the observed waveforms. We discuss this at lines 134-137.

Q6: Line 133: “A two-dimensional (2D) ULVZ model spanning 12° in the N-S direction with 50 km height and -10% δV_S predicts S and ScS data for all events A-D along this particular corridor (Figs. S8-S9).” I think many readers will find a “2D” model confusing. I understand that the authors performed both 2D and 3D numerical simulations, referring to the ULVZ as a 2D structure doesn’t make sense to me. The 2D simulations are simply taking a slice through the 3D ULVZ.

A6: We have changed this sentence as “Through the three-dimensional (3D) ULVZ structure, a two-dimensional (2D) slice spanning 12° in the N-S direction with 50 ± 20 km height and $-10\pm 2\%$ δV_S predicts S and ScS data for all events A–D along this 2D corridor. (new Figs. S8-S9)” at lines 134-137.

Q7: It seems like some important results are being relegated to the Supplemental material. For example, the results shown in Figs S1 and S2 are the first results discussed in the paper and may be useful for the readers to understand the context of the study.

A7: We have merged the old Fig. S1c-d and Fig. S2 into new Fig. 2.

REVIEWERS' COMMENTS

Reviewer #2 (Remarks to the Author):

The manuscript has been improved by the authors based on the suggestions from all the reviewers. Especially, they include an extra session on how they measure the misfit between the synthetics and observations.

Now my only concern is that I think the authors still need to expand more details in the geodynamical reasons for the NP mega-sized ULVZ to the readers. For example, the authors mentioned the viscosity of ULVZ with a smaller velocity drop might be larger than ULVZs with extreme velocity drops, then why does a more viscous NP ULVZ have a larger lateral extent than a less viscous ULVZ beneath Iceland and Marquesas? And in terms of a subvertical tilting wall of the Pacific LLVP, can it be supported by a geodynamic modeling specifically for this area, etc?

Some specific suggestions are listed below.

1. The authors claim that they measured the ScS-S differential travel time residuals with epicentral distances smaller than 85° . However, in Fig. 2, some measurements are clearly beyond this cut-off distance.
2. In Line 268, I might be wrong. I don't think He and Wen (2012) specifically identified the existence of D" discontinuity in this region. The authors should've provided some more solid citations.
3. For a high-impact journal that interests a wide range of audience, Fig. 6 (especially b and c) needs to be plotted nicer for better demonstrating the possible dynamic processes.

Reviewer #3 (Remarks to the Author):

The authors appear to have considered reviewer comments in the new version, and the manuscript has significantly improved. I recommend it be considered for publication.

Reviewer #4 (Remarks to the Author):

The authors have greatly improved the manuscript, and I am satisfied with how they have addressed all of my comments and concerns. In particular, they modified the discussion to give a more honest interpretation of their results, and added an analysis of uncertainties in their modeling. I have no further issues, and I believe that the manuscript is ready for publication.

Dear Reviewers,

We greatly appreciate your comments. The comments from reviewer #2 are followed by our responses in *blue italics*.

Q1: The manuscript has been improved by the authors based on the suggestions from all the reviewers. Especially, they include an extra session on how they measure the misfit between the synthetics and observations.

Now my only concern is that I think the authors still need to expand more details in the geodynamical reasons for the NP mega-sized ULVZ to the readers. For example, the authors mentioned the viscosity of ULVZ with a smaller velocity drop might be larger than ULVZs with extreme velocity drops, then why does a more viscous NP ULVZ have a larger lateral extent than a less viscous ULVZ beneath Iceland and Marquesas? And in terms of a subvertical tilting wall of the Pacific LLVP, can it be supported by a geodynamic modeling specifically for this area, etc?

A1: Although we mentioned the viscosity of the NP ULVZ with smaller velocity drops might be larger than ULVZs with extreme velocity drops, based on the rheology calculation in Reali et al., 2019 (Ref. 40), the effects from viscosity might be a minor factor in terms of forming ULVZs with different total mass or volume. As addressed in our paper (Line 256-264), we propose that the long-term and stable convergent mantle flow in the north Pacific is the key factor that distinguishes the formation of the NP ULVZ from other ULVZs beneath Iceland and Marquesas. We also mentioned that a strong hot upwelling may exist beneath Iceland and Samoa, which entrains and drains the ULVZ materials more efficiently despite their larger density and smaller viscosity (Line 250-255). A strong hot upwelling will generate a flow pattern that mitigates the flattening of a ULVZ due to its intrinsic density or viscosity contrast. To further clarify our points, in line 234-244, we re-organized the discussion on the density and viscosity, and focused on explaining the similar aspect ratio of the NP ULVZ and other mega-sized ULVZs beneath Iceland and Marquesas.

Thanks for the comment on the geodynamic modeling of the subvertical tilting wall of the Pacific LLVP. Although there are no direct geodynamic models for this area, we have referenced general geodynamic models regarding how a slab interacts with an LLVP edge from Ni et al., 2002 (Ref. 9), Bower et al., 2013 (Ref. 48), and a new reference Cao et al., 2021 (Ref. 51) which agrees with our observation well.

Q2: The authors claim that they measured the ScS-S differential travel time residuals with epicentral distances smaller than 85° . However, in Fig. 2, some measurements are clearly beyond this cut-off distance.

A2: There is a similar comment in an earlier review. To address that comment, we added an additional description in “Data processing procedures and travel time measurements” section in “Methods”. To avoid confusion in the main text, we further added “For event B, we measure the $\delta t_{\text{scS-S}}$ with epicentral distance extending to 90° due to the high-quality data (Fig. 2c)” at Line 70-72.

Q3: In Line 268, I might be wrong. I don't think He and Wen (2012) specifically identified the existence of D" discontinuity in this region. The authors should've provided some more solid citations.

A3: He and Wen (2012) (Ref. 24) imaged a high-velocity layer in the same region and showed a positive V_S jump in the lowermost mantle, which is typically referred to as the D" discontinuity. We rewrote this sentence as: “A D" discontinuity, characterized by 2% V_S jump at 220 km above the CMB, most likely linked to subducted slab, is also identified at the northern edge of the Pacific LLVP” at line 270-272.

Q4: For a high-impact journal that interests a wide range of audience, Fig. 6 (especially b and c) needs to be plotted nicer for better demonstrating the possible dynamic processes.

A4: We have redrawn Fig. 6 to provide a much improved illustration of the dynamic processes.